# SAM 2: SEGMENT ANYTHING IN IMAGES AND VIDEOS

**Nikhila Ravi**[*,†]    **Valentin Gabeur**[*]    **Yuan-Ting Hu**[*]    **Ronghang Hu**[*]    **Chaitanya Ryali**[*]
**Tengyu Ma**[*]    **Haitham Khedr**[*]    **Roman Rädle**[*]    **Chloe Rolland**    **Laura Gustafson**
**Eric Mintun**    **Junting Pan**    **Kalyan Vasudev Alwala**    **Nicolas Carion**    **Chao-Yuan Wu**
**Ross Girshick**    **Piotr Dollár**[†]    **Christoph Feichtenhofer**[*,†]
Meta FAIR,    `https://github.com/facebookresearch/sam2`

## ABSTRACT

We present Segment Anything Model 2 (SAM 2), a foundation model towards solving promptable visual segmentation in images and *videos*. We build a data engine, which improves model and data via user interaction, to collect the largest video segmentation dataset to date. Our model is a simple transformer architecture with streaming memory for real-time video processing. SAM 2 trained on our data provides strong performance across a wide range of tasks. In video segmentation, we observe better accuracy, using $3\times$ fewer interactions than prior approaches. In image segmentation, our model is more accurate and $6\times$ faster than the Segment Anything Model (SAM). We believe that our data, model, and insights will serve as a significant milestone for video segmentation and related perception tasks. We are releasing our main model, the dataset, an interactive demo and code.

## 1 INTRODUCTION

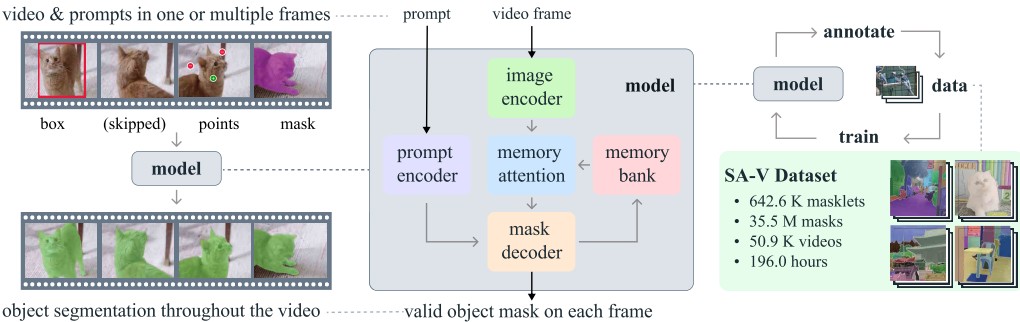

(a) **Task:** promptable visual segmentation    (b) **Model:** Segment Anything Model 2    (c) **Data:** data engine and dataset

Figure 1: We introduce the Segment Anything Model 2 (SAM 2), towards solving the promptable visual segmentation task (a) with our foundation model (b), trained on our large-scale SA-V dataset collected through our data engine (c). SAM 2 is capable of interactively segmenting regions through prompts (clicks, boxes, or masks) on one or multiple video frames by utilizing a streaming memory that stores previous prompts and predictions.

Segment Anything (SA) introduced a foundation model for promptable *segmentation* in *images* (Kirillov et al., 2023). However an image is only a static snapshot of the real world in which visual segments can exhibit complex motion, and with the rapid growth of multimedia content, a significant portion is now recorded with a temporal dimension, particularly in *video* data. Many important applications in AR/VR, robotics, autonomous vehicles, and video editing require temporal localization beyond image-level segmentation. We believe a universal visual segmentation system should be applicable to both images *and* videos.

Segmentation in video aims to determine the spatio-temporal extent of entities, which presents unique challenges beyond those in images. Entities can undergo significant changes in appearance due to motion, deformation, occlusion, lighting changes, and other factors. Videos often have lower quality than images due to camera motion, blur, and lower resolution. Further, efficient processing of a large number of frames is a key challenge. While SA successfully addresses segmentation in images, existing video segmentation models and datasets fall short in providing a comparable capability to "segment *anything* in videos".

---

[*]core contributor, [†]project lead

We introduce the Segment Anything Model 2 (SAM 2), a *unified* model for video and image segmentation (we consider an image as a single-frame video). Our work includes a task, model, and dataset (see Fig. 1).

We focus on the Promptable Visual Segmentation (PVS) *task* that generalizes image segmentation to the video domain. The task takes as input points, boxes, or masks on any frame of the video to define a segment of interest for which the spatio-temporal mask (i.e., a '*masklet*') is to be predicted. Once a masklet is predicted, it can be iteratively refined by providing prompts in additional frames.

Our *model* (§4) produces segmentation masks of the object of interest, in single images *and* across video frames. SAM 2 is equipped with a memory that stores information about the object and previous interactions, which allows it to generate masklet predictions throughout the video, and also effectively correct these based on the stored memory context of the object from previously observed frames. Our streaming architecture is a natural generalization of SAM to the video domain, processing video frames one at a time, equipped with a memory attention module to attend to the previous memories of the target object. When applied to images, the memory is empty and the model behaves like SAM.

We employ a *data engine* (§5) to generate training data by using our model in the loop with annotators to interactively annotate new and challenging data. Different from most existing video segmentation datasets, our data engine is not restricted to objects of specific categories, but instead targeted to provide training data for segmenting *any* object with a valid boundary, including parts and subparts. Compared to existing model-assisted approaches, our data engine with SAM 2 in the loop is $8.4\times$ faster at comparable quality. Our final Segment Anything Video (SA-V) dataset (§5.2) consists of 35.5M masks across 50.9K videos, $53\times$ more masks than any existing video segmentation dataset. SA-V is challenging with small objects and parts that get occluded and re-appear throughout the video. Our SA-V dataset is geographically diverse, and a fairness evaluation of SAM 2 indicates minimal performance discrepancy in video segmentation based on perceived gender, and little variance among the three perceived age groups we evaluated.

Our experiments (§6) show that SAM 2 delivers a step-change in the *video* segmentation experience. SAM 2 can produce *better* segmentation accuracy while using $3\times$ *fewer* interactions than prior approaches. Further, SAM 2 outperforms prior work in established *video* object segmentation benchmarks, under multiple evaluation settings, *and* delivers better performance compared to SAM on *image* segmentation benchmarks, while being $6\times$ faster. SAM 2 is shown to be effective across a variety of video and image distributions as observed through numerous zero-shot benchmarks including 17 for video segmentation and 37 for single-image segmentation.

We are releasing our work under permissive open licences, including the SA-V dataset, the SAM 2 model checkpoints, training code, and code for an interactive web demo.

## 2 RELATED WORK

**Image segmentation.** Segment Anything (Kirillov et al., 2023) introduces a promptable image segmentation task where the goal is to output a valid segmentation mask given an input prompt such as a bounding box or a point that refers to the object of interest. SAM trained on the SA-1B dataset allows for zero-shot segmentation which enabled its adoption to a wide range of applications. Recent work has extended SAM, e.g., by introducing a High-Quality output token to train on fine-grained masks (Ke et al., 2024), or improve SAM's efficiency (Xiong et al., 2023; Zhang et al., 2023a; Zhao et al., 2023). More broadly, SAM is used in a wide range of applications, including medical imaging (Ma et al., 2024; Deng et al., 2023; Mazurowski et al., 2023; Wu et al., 2023a), remote sensing (Chen et al., 2024; Ren et al., 2024), motion segmentation (Xie et al., 2024), and camouflaged object detection (Tang et al., 2023).

**Interactive Video Object Segmentation (iVOS).** Interactive video object segmentation has emerged as a crucial task to efficiently obtain object segmentations in videos (masklets) with user guidance, often in the form of scribbles, clicks, or bounding boxes. A few early approaches (Wang et al., 2005; Bai & Sapiro, 2007; Fan et al., 2015) deploy graph-based optimization to guide the segmentation annotation process. More recent approaches (Heo et al., 2020; Cheng et al., 2021b; Delatolas et al., 2024) often adopt a modular design, converting user inputs into a mask representation on a single frame and then propagating it to other frames.

Click-based input is easier to collect (Homayounfar et al., 2021) for interactive video segmentation. Recent works have used a combination of SAM on images with video trackers based on masks (Cheng et al., 2023b; Yang et al., 2023; Cheng et al., 2023c) or points (Rajič et al., 2023). However, these approaches have limitations: the tracker may not work for all objects, SAM may not perform well on video frames, and there is no mechanism to interactively refine a model's mistakes, other than re-annotating using SAM in each frame and restarting the tracking from there.

Our work shares a similar goal to these works to segment objects across videos interactively, and we build a strong unified model that directly takes prompts for interactive video segmentation, along with a large and diverse dataset in pursuit of solving this goal.

**Video Object Segmentation (VOS).** The VOS task begins with an object mask as input in the first frame, which must be accurately tracked throughout the video (Pont-Tuset et al., 2017). The task is referred to as "semi-supervised VOS" since the input mask can be seen as supervision signal of the object which is available only in the first frame. This task has drawn significant attention due to its relevance in applications, including video editing or robotics.

Early deep learning based approaches have often used fine-tuning on the first video frame (Caelles et al., 2016; Perazzi et al., 2016; Yoon et al., 2017; Maninis et al., 2017; Hu et al., 2018a; Bhat et al., 2020; Robinson et al., 2020) or on all frames (Voigtlaender & Leibe, 2017) to adapt the model to the target object. Faster inference has been achieved with offline-trained models, conditioned either only on the first frame (Hu et al., 2018b; Chen et al., 2018), or also integrating the previous frame (Oh et al., 2018; Yang et al., 2018; 2020). This multi-conditioning has been extended to all frames with RNNs (Tokmakov et al., 2017; Xu et al., 2018a) and transformers (Oh et al., 2019; Cheng et al., 2021a; Li et al., 2022a; Yang et al., 2021b; 2024; Cheng & Schwing, 2022; Yang & Yang, 2022; Wang et al., 2022; Cheng et al., 2023a; Goyal et al., 2023; Zhang et al., 2023b; Wu et al., 2023b).

Semi-supervised VOS can be seen as a special case of our Promptable Visual Segmentation (PVS) task, with only a mask prompt in the first video frame. Notably, annotating the required high-quality object mask in the first frame in VOS is practically challenging and time-consuming for inference.

**Video segmentation datasets.** Many datasets have been proposed for VOS. Early VOS datasets (Prest et al., 2012; Li et al., 2013; Ochs et al., 2014; Fan et al., 2015), such as DAVIS (Pont-Tuset et al., 2017; Caelles et al., 2019), include high-quality annotations but their size limits deep-learning based approaches. YouTube-VOS (Xu et al., 2018b) is the first large-scale dataset for VOS. As algorithms became better and benchmark performance started to saturate, researchers have looked at increasing the difficulty of the VOS task by specifically focusing on occlusions (Qi et al., 2022; Ding et al., 2023), long videos (Hong et al., 2023; 2024), extreme transformations (Tokmakov et al., 2022), object diversity (Wang et al., 2021b; 2023) or scene diversity (Athar et al., 2022; Xu et al., 2023).

We find that current video segmentation datasets lack sufficient coverage to achieve the capability of "segmenting *anything* in videos". Their annotations typically cover entire objects (not parts) and datasets are often centered around specific object classes, such as people, vehicles, and animals. In comparison to these datasets, our released SA-V dataset not only focuses on whole objects but also extensively covers object parts and contains over an order of magnitude more masks.

## 3 TASK: PROMPTABLE VISUAL SEGMENTATION

Our PVS task allows providing prompts to the model on *any* frame of a video. Prompts can be positive/negative clicks, boxes, or masks, either to define an object to segment or to refine a model-predicted one. To provide an interactive experience, upon receiving a prompt on a specific frame, the model should immediately respond with a valid segmentation mask of the object on this frame. After receiving initial prompts (either on the same frame or different frames), the model should propagate these prompts to obtain the masklet of the object *across the entire video*, localizing the segmentation mask of the target on every video frame. Additional prompts can be provided to the model on *any* frame to refine the segment throughout the video (example in Fig. 2). For details on the task, see §B.

SAM 2 (§4) is applied as a data collection tool to the PVS task for building our SA-V dataset (§5). We evaluate the model (§6) by simulating interactive video segmentation scenarios across multiple frames, in the conventional semi-supervised VOS setting where annotations are limited to the first frame, and for image segmentation on the SA benchmarks.

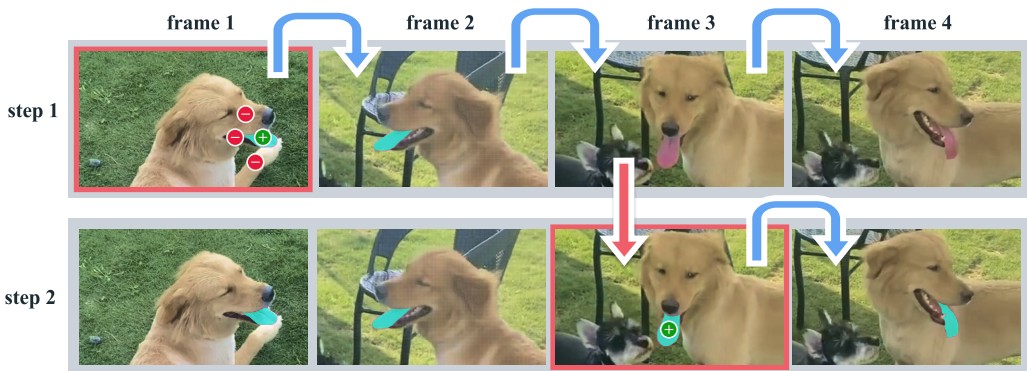

Figure 2: Interactive segmentation with SAM 2. Step 1 (selection): we prompt SAM 2 in frame 1 to obtain the segment of the target object (the tongue). Green/red dots indicate positive/negative prompts respectively. SAM 2 automatically propagates the segment to the following frames (blue arrows) to form a *masklet*. If SAM 2 loses the object (after frame 2), we can correct the masklet by providing an additional prompt in a new frame (red arrow). Step 2 (refinement): a single click in frame 3 is sufficient to recover the object and propagate it to obtain the correct masklet. A decoupled SAM + video tracker approach would require several clicks in frame 3 (as in frame 1) to correctly re-annotate the object as the segmentation is restarted from scratch. With SAM 2's memory, a single click can recover the tongue.

## 4 MODEL

SAM 2 (Fig. 3) can be seen as a generalization of SAM to the video (and image) domain, taking point, box, and mask prompts on *individual* frames to define the spatial extent of the object to be segmented spatio-temporally. Spatially, the model behaves similarly to SAM. A promptable and *light-weight* mask decoder takes an image embedding and prompts (if any) and outputs a segmentation mask for the frame. Prompts can be *iteratively* added on a frame in order to *refine* the masks.

The frame embedding used by the SAM 2 decoder is not directly from an image encoder and is instead *conditioned* on *memories* of past predictions and *prompted frames*. It is possible for prompted frames to also come "from the future" relative to the current frame. Memories of frames are created by the *memory encoder* based on the current prediction and placed in a *memory bank* for use in subsequent frames. The *memory attention* operation takes the per-frame embedding from the image encoder and conditions it on the memory bank, before the mask decoder ingests it to form a prediction.

We describe individual components and training below and provide more details in Appendix D.

**Image encoder.** For real-time processing of arbitrarily long videos, we take a streaming approach, consuming video frames as they become available. The image encoder is only run *once* for the entire interaction and its role is to provide unconditioned tokens (feature embeddings) representing each frame. We use an MAE (He et al., 2022) pre-trained Hiera (Ryali et al., 2023; Bolya et al., 2023) image encoder, which is *hierarchical*, allowing us to use multiscale features during decoding.

**Memory attention.** The role of memory attention is to *condition* the current frame features on the past frames features and predictions as well as on any new prompts. We stack $L$ transformer blocks, the first one taking the image encoding from the current frame as input. Each block performs self-attention, followed by cross-attention to memories of (prompted/unprompted) frames and object pointers (see below), stored in a *memory bank* (see below), followed by an MLP. We use *vanilla* attention operations for self- and cross-attention, allowing us to benefit from recent developments in efficient attention kernels (Dao, 2023).

**Prompt encoder and mask decoder.** Our prompt encoder is identical to SAM's and can be prompted by clicks (positive or negative), boxes, or masks to define the extent of the object in a given frame. Sparse prompts are represented by positional encodings summed with learned embeddings for each prompt type, while masks are embedded using convolutions and summed with the frame embedding.

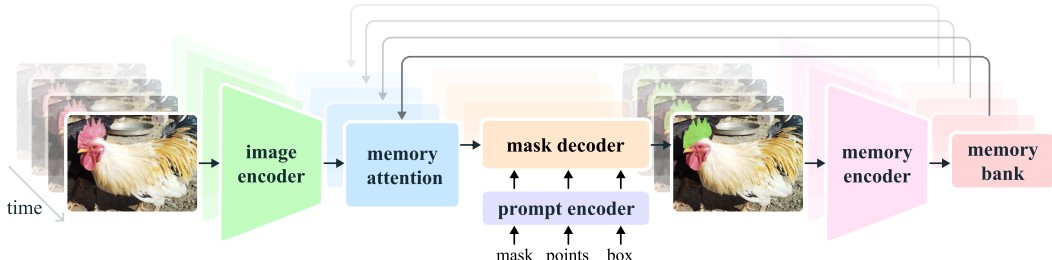

Figure 3: The SAM 2 architecture. For a given frame, the segmentation prediction is conditioned on the current prompt *and/or* on previously observed memories. Videos are processed in a *streaming* fashion with frames being consumed one at a time by the image encoder, and cross-attended to memories of the target object from previous frames. The mask decoder, which optionally also takes input prompts, predicts the segmentation mask for that frame. Finally, a memory encoder transforms the prediction and image encoder embeddings (not shown in the figure) for use in future frames.

Our decoder design largely follows SAM. We stack "two-way" transformer blocks that update prompt and frame embeddings. As in SAM, for ambiguous prompts (i.e., a single click) where there may be multiple compatible target masks, we predict *multiple* masks. This design is important to ensure that the model outputs *valid* masks. In *video*, where ambiguity can extend *across* video frames, the model predicts multiple masks on each frame. If no follow-up prompts resolve the ambiguity, the model only propagates the mask with the highest predicted IoU for the current frame.

Unlike SAM where there is *always* a valid object to segment given a positive prompt, in the PVS task it is possible for *no* valid object to exist on some frames (e.g. due to occlusion). To support this new output mode, we add an additional head that predicts whether the object of interest is present on the current frame. Another novelty are skip connections from our hierarchical image encoder (bypassing the memory attention) to incorporate *high-resolution* embeddings for mask decoding (see §D).

**Memory encoder.** The memory encoder generates a memory by downsampling the output mask using a convolutional module and summing it element-wise with the *unconditioned frame embedding* from the image-encoder (not shown in Fig. 3), followed by light-weight convolutional layers to fuse the information.

**Memory bank.** The memory bank retains information about past predictions for the target object in the video by maintaining a FIFO queue of memories of up to $N$ recent frames and stores information from prompts in a FIFO queue of up to $M$ prompted frames. For instance, in the VOS task where the initial mask is the only prompt, the memory bank consistently retains the first frame's memory along with memories of up to $N$ recent (unprompted) frames. Both sets of memories are stored as *spatial* feature maps.

In addition to the spatial memory, we store a list of *object pointers* as lightweight vectors for high-level semantic information of the object to segment, based on mask decoder output tokens of each frame. Our memory attention cross-attends to both spatial memory features and these object pointers.

We embed temporal position information into the memories of $N$ recent frames, allowing the model to represent short-term object motion, but not into those of prompted frames, because the training signal from prompted frames is sparser and it is more difficult to generalize to the inference setting where prompted frames may come from a very different temporal range than seen during training.

**Training.** The model is trained *jointly* on image and video data. Similar to previous work (Kirillov et al., 2023; Sofiiuk et al., 2022), we *simulate* interactive prompting of the model. We sample sequences of 8 frames and randomly select up to 2 frames to prompt and probabilistically receive *corrective* clicks which are sampled using the ground-truth masklet and model predictions during training. The training task is to sequentially (and "interactively") predict the ground-truth masklet. Initial prompts to the model can be the ground-truth mask with probability $0.5$, a positive click sampled from the ground-truth mask with probability $0.25$, or a bounding box input with probability $0.25$. See §D for more details.

## 5 DATA

To develop the capability to "segment anything" in video, we built a data engine to collect a large and diverse video segmentation dataset. We employ an interactive model in the loop setup with human annotators. Similar to Kirillov et al. (2023), we do not impose semantic constraints on the annotated masklets, and focus on both whole objects (e.g., a person) and parts (e.g., a person's hat). Our data engine went through three phases, each categorized based on the level of model assistance provided to annotators. Next, we describe each data engine phase and our SA-V dataset.

### 5.1 DATA ENGINE

**Phase 1: SAM per frame.** The initial phase used the image-based interactive SAM (Kirillov et al., 2023) to assist human annotation. Annotators are tasked with annotating the mask of a target object in every frame of the video at 6 frames per second (FPS) using SAM, and pixel-precise manual editing tools such as a "brush" and "eraser". There is no tracking model involved to assist with the temporal propagation of masks to other frames. As this is a per-frame method, and all frames require mask annotation from scratch, the process is slow, with an average annotation time of 37.8 seconds per frame in our experiment. However, this yields high-quality *spatial* annotations per frame. In this phase, we collected 16K masklets across 1.4K videos. We further use this approach to annotate our SA-V val and test sets to mitigate potential biases of SAM 2 during evaluation.

**Phase 2: SAM + SAM 2 Mask.** The second phase added SAM 2 into the loop, where SAM 2 only accepted *masks* as prompts. We refer to this version as SAM 2 Mask. Annotators used SAM and other tools as in Phase 1 to generate *spatial* masks in the first frame, and then use SAM 2 Mask to *temporally* propagate the annotated mask to other frames to get the full spatio-temporal masklets. At any subsequent video frame, annotators can *spatially* modify the predictions made by SAM 2 Mask by annotating a mask from scratch with SAM, a "brush" and/or "eraser", and re-propagate with SAM 2 Mask, repeating this process until the masklet is correct. SAM 2 Mask was initially trained on the Phase 1 data and publicly available datasets. During Phase 2, we re-trained and updated SAM 2 Mask in the annotation loop twice using the collected data. In Phase 2, we collected 63.5K masklets. The annotation time went down to 7.4 s/frame, a ~**5.1x** speed up over Phase 1.

Despite an improvement in annotation time, this approach requires annotating masks in intermediate frames from scratch without previous memory. We then advanced to develop the fully-featured SAM 2, capable of both interactive segmentation and mask propagation in a *unified* model.

**Phase 3: SAM 2.** In the final phase, we utilize the fully-featured SAM 2, which accepts various types of prompts, including points and masks. SAM 2 benefits from memories of objects across the temporal dimension to generate mask predictions. This means annotators only need to provide occasional *refinement* clicks to SAM 2 to edit the predicted masklets in intermediate frames, as opposed to annotating from scratch with a spatial SAM which has no such memory context. During Phase 3, we re-trained and updated SAM 2 using the collected annotations five times. With SAM 2 in the loop, the annotation time per frame went down to 4.5 seconds, a ~**8.4x** speed up over Phase 1. In Phase 3, we collected 197.0K masklets.

**Quality verification.** To uphold a high standard for annotation, we introduce a verification step. A separate set of annotators are tasked with verifying the quality of each annotated masklet as "satisfactory" (correctly and consistently tracking the target object across all frames) or "unsatisfactory" (target object is *well defined* with a clear boundary but the masklet is not correct or consistent). Unsatisfactory masklets were sent back to the annotation pipeline for refinement. Any masklets tracking *not well defined* objects were rejected entirely.

**Auto masklet generation.** Ensuring diversity in annotation is important to enable the *anything capability* of our model. As human annotators might typically focus more on salient objects, we augment the annotations with automatically generated masklets (referred to as "Auto"). This serves a dual purpose of increasing the coverage of annotations and helping identify model failure cases. To generate auto masklets, we prompt SAM 2 with a regular grid of points in the first frame and generate candidate masklets. These are then sent to the masklet verification step for filtering. Automatic masklets tagged as "satisfactory" are added to the SA-V dataset. Masklets identified as "unsatisfactory"

| | Model in the Loop | Time per Frame | Edited Frames | Clicks per Clicked Frame | Phase 1 Mask Alignment Score (IoU>0.75) | | | |
|---|---|---|---|---|---|---|---|---|
| | | | | | All | Small | Medium | Large |
| Phase 1 | SAM only | 37.8 s | 100.00 % | 4.80 | - | - | - | - |
| Phase 2 | SAM + SAM 2 Mask | 7.4 s | 23.25 % | 3.61 | 86.4 % | 71.3 % | 80.4 % | 97.9 % |
| Phase 3 | **SAM 2** | **4.5 s** | **19.04 %** | **2.68** | **89.1** % | **72.8** % | **81.8** % | **100.0** % |

Table 1: Evolution of data engine phases showing the average annotation time per frame, the average percent of edited frames per masklet, the number of manual clicks per clicked frame, and Mask Alignment to Phase 1 by mask size.

(i.e., model failure cases) are sampled and presented to annotators to refine with SAM 2 in the loop (Phase 3 of the data engine). These automatic masklets cover large salient central objects but also objects of varying sizes and positions in the background.

**Analysis.** Table 1 shows a comparison of the annotation protocol in each data engine phase through a controlled experiment (details in §E.2.2). We compare the average annotation time per frame, the average percentage of manually edited frames per masklet, and the average number of clicks per clicked frame. For quality evaluation, we define the *Phase 1 Mask Alignment Score* as the percentage of masks whose IoU compared to the corresponding masks in Phase 1 exceeds 0.75. Phase 1 data is chosen as a reference as it has per-frame high quality manual annotations. Phase 3 with SAM 2 in the loop leads to increased efficiency and comparable quality: it is 8.4× faster than Phase 1, has the lowest edited frame percentage and clicks per frame, and results in better alignment.

In Table 2, we show the performance comparison of SAM 2 trained on the available data at the end of each phase keeping the number of iterations *fixed*, therefore measuring solely the impact of the additional data. We evaluate on our own SA-V val set and also on 9 zero-shot benchmarks (see §F.1 for details) using the standard $\mathcal{J}\&\mathcal{F}$ accuracy metric (the higher the better) when prompting with 3-clicks on the first frame. We note a consistent improvement after iteratively including the data from each phase, not only on the in-domain SA-V val set, but also on the 9 zero-shot benchmarks.

| Training data | SA-V val | 9 zero-shot |
|---|---|---|
| VOS + SA-1B | 50.0 | 62.5 |
| + Phase 1 | 53.0 | 66.9 |
| + Phase 2 | 58.8 | 70.9 |
| + Phase 3 | 62.5 | 71.2 |
| + Auto | **63.2** | **71.5** |

Table 2: Segmentation accuracy ($\mathcal{J}\&\mathcal{F}$ metric) improvement from adding data from each data engine phase. "VOS" is a set of video object segmentation datasets. Details are in §F.

### 5.2 SA-V DATASET

The SA-V dataset collected with our data engine comprises 50.9K videos with 642.6K masklets. In Table 3 we compare the SA-V composition to common VOS datasets across the number of videos, masklets, and masks. Notably, the number of annotated masks is 53× (15× without auto) larger than any existing VOS dataset, providing a substantial resource for future work. We are releasing SA-V under a permissive license.

**Videos.** We collected a new set of 50.9K videos captured by crowdworkers. Videos comprise 54% indoor and 46% outdoor scenes with an average duration of 14 seconds. Videos feature "*in-the-wild*" diverse environments, and cover various everyday scenarios.

**Masklets.** The annotations comprise 190.9K manual masklet annotations and 451.7K automatic masklets collected using our data engine. Example videos with masklets overlaid (manual and automatic) are shown in Fig. 4. SA-V has 53× (15× without auto annotations) more masks than the largest VOS dataset. The disappearance rate (Ding et al., 2023) in SA-V Manual (the percentage of annotated masklets that disappear in at least one frame and then re-appear) is 42.5%, competitive among existing datasets.

**SA-V training, validation and test splits.** We split SA-V based on the video authors (and their geographic locations) to ensure minimal overlap of similar objects. To create SA-V val and SA-V test sets, we focus on challenging scenarios in selecting videos, and ask annotators to identify *challenging targets* that are fast-moving, have complex occlusions by other objects as well as disappearance/re-appearance patterns. These targets were annotated at 6 FPS using the data engine Phase 1 setup in §5.1. There are 293 masklets and 155 videos in the SA-V val split, and 278 masklets and 150 videos in the SA-V test split.

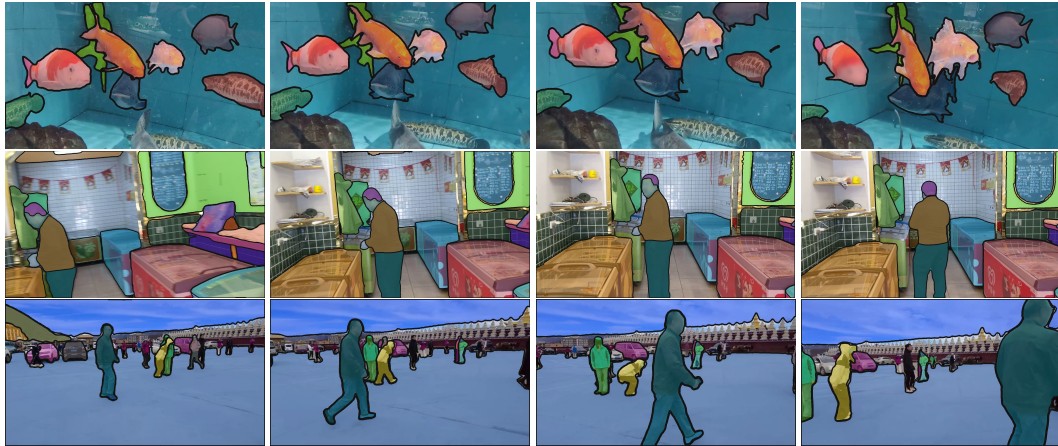

Figure 4: Example videos from the SA-V dataset with masklets overlaid (manual and automatic). Each masklet has a unique color, and each row represents frames from one video, with 1 second between them. More examples in Fig. 11.

| | #Videos | Duration | #Masklets | #Masks | #Frames | Disapp. Rate |
|---|---|---|---|---|---|---|
| DAVIS 2017 (Pont-Tuset et al., 2017) | 0.2K | 0.1 hr | 0.4K | 27.1K | 10.7K | 16.1 % |
| YouTube-VOS (Xu et al., 2018b) | 4.5K | 5.6 hr | 8.6K | 197.3K | 123.3K | 13.0 % |
| UVO-dense (Wang et al., 2021b) | 1.0K | 0.9 hr | 10.2K | 667.1K | 68.3K | 9.2 % |
| VOST (Tokmakov et al., 2022) | 0.7K | 4.2 hr | 1.5K | 175.0K | 75.5K | 41.7 % |
| BURST (Athar et al., 2022) | 2.9K | 28.9 hr | 16.1K | 600.2K | 195.7K | 37.7 % |
| MOSE (Ding et al., 2023) | 2.1K | 7.4 hr | 5.2K | 431.7K | 638.8K | 41.5 % |
| Internal | 62.9K | 281.8 hr | 69.6K | 5.4M | 6.0M | 36.4 % |
| SA-V Manual | 50.9K | 196.0 hr | 190.9K | 10.0M | 4.2M | 42.5 % |
| SA-V Manual+Auto | 50.9K | 196.0 hr | 642.6K | 35.5M | 4.2M | 27.7 % |

Table 3: Comparison of our datasets with open source VOS datasets in terms of number of videos, duration, number of masklets, masks, frames, and disappearance rate. SA-V Manual contains only manually annotated labels. SA-V Manual+Auto combines manually annotated labels with automatically generated masklets.

**Internal dataset.** We also used internally available licensed video data to further augment our training set. Our internal dataset is comprised of 62.9K videos and 69.6K masklets annotated in Phase 2 and Phase 3 (see §5.1) for training, and 96 videos and 189 masklets annotated using Phase 1 for testing (Internal-test).

See Appendix E for more details on the data engine and SA-V dataset, including a fairness evaluation.

## 6 ZERO-SHOT EXPERIMENTS

Here, we compare SAM 2 with previous work on zero-shot video and image tasks. We report the standard $\mathcal{J}\&\mathcal{F}$ metric (Pont-Tuset et al., 2017) for video and mIoU metric for image tasks. Unless otherwise mentioned, the results in this section follow our default setup using Hiera-B+ image encoder with a resolution of 1024 and trained on the full combination of datasets, i.e., SAM 2 (Hiera-B+) in Table 6 (see also §D.2 for details).

### 6.1 PROMPTABLE VIDEO SEGMENTATION

We first evaluate promptable video segmentation, which involves simulating an interactive setting that resembles the user experience. We have two settings, *offline* evaluation, where multiple passes are made through a video to select frames to interact with based on the largest model error, and *online* evaluation, where the frames are annotated in a single forward pass through the video. These evaluations are conducted on 9 densely annotated zero-shot video datasets using $N_{\text{click}} = 3$ clicks per frame (see §F.1 for details).

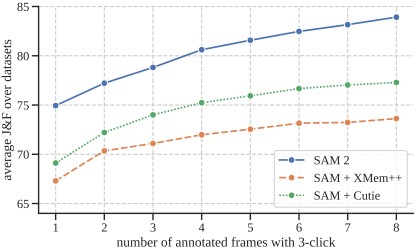 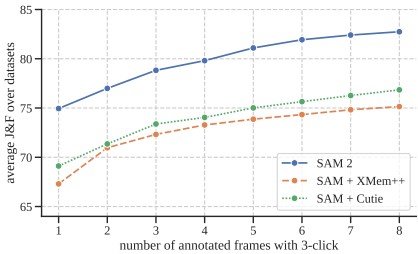

(a) *offline* average $\mathcal{J}\&\mathcal{F}$ across datasets (3-click)  (b) *online* average $\mathcal{J}\&\mathcal{F}$ across datasets (3-click)

Figure 5: Zero-shot accuracy over 9 datasets in interactive offline and online evaluation settings.

We create two strong baselines, SAM+XMem++ and SAM+Cutie, based on two state-of-the-art models for video object segmentation, XMem++ (Bekuzarov et al., 2023) and Cutie (Cheng et al., 2023a). We use XMem++ to generate a video segmentation based on mask inputs on one or multiple frames. SAM is used to provide an initial mask or to refine an output (by feeding the current segmentation as a mask prompt to SAM). For the SAM+Cutie baseline, we modify Cutie to allow taking mask inputs on multiple frames.

In Fig. 5, we report the average $\mathcal{J}\&\mathcal{F}$ accuracy over $N_{\text{frame}} = 1, \ldots, 8$ interacted frames. SAM 2 outperforms SAM+XMem++ and SAM+Cutie for both offline and online evaluation settings. Across all 9 datasets (see per-dataset results in §F.1), SAM 2 dominates both methods, generating high-quality video segmentation from a few clicks while allowing continued refinement with prompts. Overall, SAM 2 can generate better segmentation accuracy, with $>3\times$ fewer interactions.

## 6.2 Semi-supervised video object segmentation

| Method | 1-click | 3-click | 5-click | bounding box | ground-truth mask[‡] |
|---|---|---|---|---|---|
| SAM+XMem++ | 56.9 | 68.4 | 70.6 | 67.6 | 72.7 |
| SAM+Cutie | 56.7 | 70.1 | 72.2 | 69.4 | 74.1 |
| **SAM 2** | **64.7** | **75.3** | **77.6** | **74.4** | **79.3** |

Table 4: Zero-shot accuracy across 17 video datasets using different prompts. We report average accuracy for each type of prompt (1, 3 or 5 clicks, bounding boxes, or ground-truth masks) in the first video frame ([‡]: this case directly uses masks as inputs into XMem++ or Cutie without SAM).

We evaluate the semi-supervised video object segmentation (VOS) setting (Pont-Tuset et al., 2017) with click, box, or mask prompts *only* on the *first frame* of the video. When using click prompts, we interactively sample either 1, 3 or 5 clicks on the first video frame.

Similar to the interactive setting in §6.1, we compare to XMem++ and Cutie, using SAM for click and box prompts, and in their default setting when using mask prompts. We report the standard $\mathcal{J}\&\mathcal{F}$ accuracy (Pont-Tuset et al., 2017), except for on VOST (Tokmakov et al., 2022), where we report the $\mathcal{J}$ metric following its protocol. The results are in Table 4. SAM 2 outperforms both methods on the 17 datasets. The results underline that SAM 2 also excels at the conventional, non-interactive VOS task with mask input, for which these other works are specifically designed. Details are in §F.1.3.

## 6.3 Image segmentation

We evaluate SAM 2 on the Segment Anything task across 37 zero-shot datasets, including 23 datasets previously used by SAM for evaluation. 1-click and 5-click mIoUs are reported in Table 5 and we show the average mIoU by dataset domain and model speed in frames per second (FPS) on a single A100 GPU.

The first column (SA-23 All) shows accuracy on the 23 datasets from SAM. SAM 2 achieves higher accuracy (58.9 mIoU with 1 click) than SAM (58.1 mIoU with 1 click), *without* using any extra data and while being **6× faster**. This can be mainly attributed to the smaller but more effective Hiera image encoder in SAM 2.

The bottom row shows how training on our SA-1B and video data mix can further improve accuracy to 61.4% on average on the 23 datasets. We also see *exceptional* gains on the video benchmarks from SA-23 (video datasets are evaluated as images, identical to Kirillov et al. (2023)), and the 14 new video datasets we added. More detailed results including a breakdown by dataset are in §F.4.

| Model | Data | 1 (5) click mIoU | | | | |
|---|---|---|---|---|---|---|
| | | SA-23 All | SA-23 Image | SA-23 Video | 14 new Video | FPS |
| SAM | SA-1B | 58.1 (81.3) | 60.8 (82.1) | 54.5 (80.3) | 59.1 (83.4) | 21.7 |
| **SAM 2** | SA-1B | 58.9 (81.7) | 60.8 (82.1) | 56.4 (81.2) | 56.6 (83.7) | **130.1** |
| **SAM 2** | our mix | **61.9 (83.5)** | **63.3 (83.8)** | **60.1 (83.2)** | **69.6 (85.8)** | **130.1** |

Table 5: Zero-shot accuracy on the Segment Anything (SA) task across 37 datasets. The table shows the average 1- and 5-click mIoU of SAM 2 compared to SAM by domains (image/video). We report the average metrics on the 23 datasets used by SAM (SA-23) and the average across 14 additional zero-shot video datasets (as detailed in §F.4).

| Method | $\mathcal{J}\&\mathcal{F}$ | | | | | $\mathcal{G}$ |
|---|---|---|---|---|---|---|
| | MOSE val | DAVIS 2017 val | LVOS val | SA-V val | SA-V test | YTVOS 2019 val |
| STCN (Cheng et al., 2021a) | 52.5 | 85.4 | - | 61.0 | 62.5 | 82.7 |
| SwinB-AOT (Yang et al., 2021b) | 59.4 | 85.4 | - | 51.1 | 50.3 | 84.5 |
| SwinB-DeAOT (Yang & Yang, 2022) | 59.9 | 86.2 | - | 61.4 | 61.8 | 86.1 |
| RDE (Li et al., 2022a) | 46.8 | 84.2 | - | 51.8 | 53.9 | 81.9 |
| XMem (Cheng & Schwing, 2022) | 59.6 | 86.0 | - | 60.1 | 62.3 | 85.6 |
| SimVOS-B (Wu et al., 2023b) | - | 88.0 | - | 44.2 | 44.1 | 84.2 |
| JointFormer (Zhang et al., 2023b) | - | 90.1 | - | - | - | 87.4 |
| ISVOS (Wang et al., 2022) | - | 88.2 | - | - | - | 86.3 |
| DEVA (Cheng et al., 2023b) | 66.0 | 87.0 | 55.9 | 55.4 | 56.2 | 85.4 |
| Cutie-base (Cheng et al., 2023a) | 69.9 | 87.9 | 66.0 | 60.7 | 62.7 | 87.0 |
| Cutie-base+ (Cheng et al., 2023a) | 71.7 | 88.1 | - | 61.3 | 62.8 | 87.5 |
| SAM 2 (Hiera-B+) | 76.6 | 90.2 | **78.0** | 76.8 | 77.0 | 88.6 |
| SAM 2 (Hiera-L) | **77.9** | **90.7** | **78.0** | **77.9** | **78.4** | **89.3** |

Table 6: VOS comparison to prior work. SAM 2 performs well in accuracy ($\mathcal{J}\&\mathcal{F}$, $\mathcal{G}$) for video segmentation based on first-frame ground-truth mask prompts. SAM 2 performs significantly better on SA-V val/test.

# 7 COMPARISON TO STATE-OF-THE-ART IN SEMI-SUPERVISED VOS

Our primary focus is on the general, interactive PVS task, but we also address the specific semi-supervised VOS setting (where the prompt is a ground-truth mask on the first frame), as it is a historically common protocol. We evaluate two versions of SAM 2 with varying image encoder sizes (Hiera-B+/-L) with different speed-vs-accuracy tradeoffs. We measure frames per second (FPS) on a single A100 GPU using a batch-size of one. SAM 2 based on Hiera-B+ and Hiera-L runs at real-time speeds of 43.8 and 30.2 FPS, respectively.

We present a comparison with existing state-of-the-art in Table 6, reporting accuracy using standard protocols. SAM 2 shows significant improvement over the best existing methods. We observe that using a larger image encoder brings significant accuracy gains across the board.

We also evaluate existing work on the SA-V val and test sets which measure performance for open-world segments of "any" object class. When comparing on this benchmark, we see that most previous methods peak at around the same accuracy. The best performance on SA-V val and SA-V test for prior work is significantly lower demonstrating the gap to a "segment *anything* in videos" capability. Finally, we see that SAM 2 also brings notable gains in long-term video object segmentation as observed in the LVOS benchmark result. For data and model ablations, see §A.

# 8 CONCLUSION

We present a natural evolution of Segment Anything into the video domain, based on three key aspects: (i) extending the promptable segmentation task to video, (ii) equipping the SAM architecture to use memory when applied to video, and (iii) the diverse SA-V dataset for training and benchmarking video segmentation. We believe SAM 2 marks a significant advancement in visual perception, positioning our contributions as milestones that will propel further research and applications.

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

APPENDIX

**Table of contents:**

## A    DATA AND MODEL ABLATIONS

This section presents ablations that informed the design decisions for SAM 2. We evaluate on SA-V val, Internal-test, our MOSE development set ("MOSE dev") which contains 200 randomly-sampled videos from the MOSE training split, excluded from our training data and the average over 9 zero-shot video datasets. As the metric for comparison, we report $\mathcal{J}\&\mathcal{F}$ under 3-click input on the first frame as a balance between the 1-click regime and the VOS-style mask prompts. Additionally, we report the average 1-click mIoU on the 23-dataset benchmark used by SAM for the SA task on images. Unless otherwise specified, we perform our ablations at $512^2$ spatial resolution, trained with SA-V manual and a 10% subset of SA-1B. Additional details are in §D.2.

### A.1    DATA ABLATIONS

| | Training data | | | | $\mathcal{J}\&\mathcal{F}$ | | | | mIoU |
|---|---|---|---|---|---|---|---|---|---|
| | VOS | Internal | SA-V | SA-1B | SA-V val | Internal-test | MOSE dev | 9 zero-shot | SA-23 |
| 1 | ✓ | | | | 48.1 | 60.2 | 76.9 | 59.7 | 45.4 |
| 2 | | ✓ | | | 57.0 | 72.2 | 70.6 | 70.0 | 54.4 |
| 3 | | | ✓ | | 63.0 | 72.6 | 72.8 | 69.7 | 53.0 |
| 4 | | | ✓ | ✓ | 62.9 | 73.2 | 73.6 | 69.7 | _58.6_ |
| 5 | | ✓ | ✓ | | 63.0 | 73.2 | 73.3 | 70.9 | 55.8 |
| 6 | | ✓ | ✓ | ✓ | **63.6** | **75.0** | 74.4 | _71.6_ | _58.6_ |
| 7 | ✓ | | | ✓ | 50.0 | 63.2 | 77.6 | 62.5 | 54.8 |
| 8 | ✓ | ✓ | | | 54.9 | 71.5 | 77.9 | 70.6 | 55.1 |
| 9 | ✓ | | ✓ | | 61.6 | 72.8 | 78.3 | 69.9 | 51.0 |
| 10 | ✓ | | ✓ | ✓ | 62.2 | 74.1 | _78.5_ | 70.3 | 57.3 |
| 11 | ✓ | ✓ | ✓ | | 61.8 | _74.4_ | _78.5_ | **71.8** | 55.7 |
| 12 | ✓ | ✓ | ✓ | ✓ | _63.1_ | 73.7 | **79.0** | _71.6_ | **58.9** |

Table 7: We train our model on different data mixtures including VOS (DAVIS, MOSE, YouTubeVOS), and subsets of Internal-train, SA-V, and SA-1B. We report the $\mathcal{J}\&\mathcal{F}$ accuracy when prompted with 3 clicks in the first frame on SA-V val and Internal-test, MOSE, and 9 zero-shot datasets, and the average 1-click mIoU on SA-23 datasets.

**Data mix ablation.** In Table 7, we compare the accuracy of SAM 2 when trained on different data mixtures. We pre-train on SA-1B and then train a separate model for each setting. We fix the number of iterations (200k) and batch size (128) with *only* the training data changing between experiments. We report accuracy on our SA-V val and Internal set, MOSE, 9 zero-shot video benchmarks, and the SA-23 tasks (§6.3).

Row 1 shows that a model purely trained on existing VOS datasets (DAVIS, MOSE, YouTubeVOS) performs well on the in-domain MOSE dev, but *poorly on all the others* including the 9 zero-shot VOS datasets (59.7 $\mathcal{J}\&\mathcal{F}$). We observe tremendous benefit from adding our data engine data into the training mix, including **+12.1**% average performance improvement on 9 zero-shot datasets (row 11 vs 1). This can be attributed to the limited coverage and size of VOS datasets. Adding SA-1B images improves the performance on the image segmentation task (rows 3 vs 4, 5 vs 6, 9 vs 10, 11 vs

12) without degrading the VOS capability. Training only on SA-V and SA-1B (row 4) is enough to obtain strong performance on all benchmarks except for MOSE (specific object categories). Overall, we obtain the best results when mixing all datasets: VOS, SA-1B, and our data engine data (row 12).

**Data quantity ablation.** Next, we study the effect of scaling training data. SAM 2 is pre-trained on SA-1B before training on varying sizes of SA-V. We report average $\mathcal{J}\&\mathcal{F}$ score (when prompted with 3 clicks in the first frame) over 3 benchmarks: SA-V val, zero-shot, and MOSE dev. Fig. 6 shows a consistent power law relationship between the quantity of training data and the video segmentation accuracy on all benchmarks.

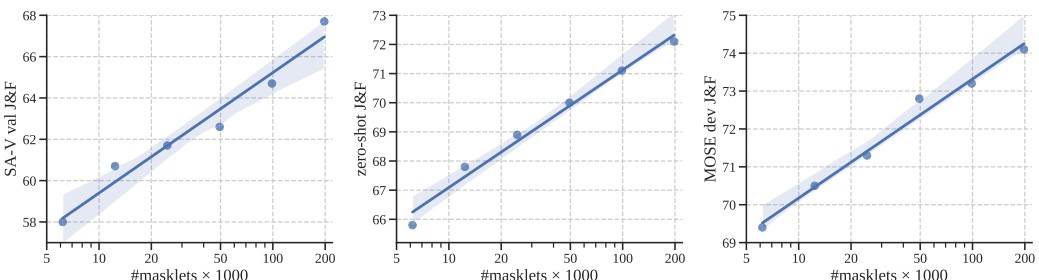

Figure 6: SAM 2 accuracy as a function of the SA-V quantity. We report $\mathcal{J}\&\mathcal{F}$ accuracy for 3-click prompts in the first frame on SA-V val (left), 9 zero-shot datasets (center), and MOSE dev (right).

**Data quality ablation.** In Table 8, we experiment with filtering strategies for quality. We subsample 50k masklets from SA-V, either randomly or by taking the masklets that have been *edited the most* by annotators. Filtering based on the number of edited frames leads to strong performance using just 25% of the data, and outperforms random sampling, but is worse than using all 190k SA-V masklets.

| | $\mathcal{J}\&\mathcal{F}$ | | | | mIoU |
|---|---|---|---|---|---|
| Setting | SA-V val | Internal-test | MOSE dev | 9 zero-shot | SA-23 |
| SA-1B + SA-V 50k random | 63.7 | 70.3 | 72.3 | 68.7 | 59.1 |
| SA-1B + SA-V 50k most edited | 66.2 | 73.0 | 72.5 | 69.2 | 58.6 |
| SA-1B + SA-V | **69.9** | **73.8** | **73.9** | **70.8** | **59.8** |

Table 8: We train our model on different subsets of our SA-V Manual data: 50k randomly sampled masklets, 50k masklets with the most edited frames, and the full SA-V dataset (190k masklets).

## A.2 MODEL ARCHITECTURE ABLATIONS

In this section, we present model ablations that guided design decisions, conducted under a smaller model setup with 512 input resolution by default. For each ablation setting, we report segmentation accuracy for video ($\mathcal{J}\&\mathcal{F}$) and image (mIoU) tasks, and its relative video segmentation speed (the maximum inference throughput relative to the ablation default setup in gray ). We find design choices for image and video components to be largely decoupled – this can be attributed to our modular design and training strategy.

### A.2.1 CAPACITY ABLATIONS

**Input size.** During training, we sample sequences of frames of fixed resolution and fixed length (here denoted by # frames). We ablate their impact in Tables 9a, 9b. A higher resolution leads to significant improvements across image and video tasks, and we use a spatial input resolution of $1024^2$ in our final model. Increasing the number of frames brings notable gains on video benchmarks and we use a default of 8 to balance speed and accuracy.

**Memory size.** Increasing the (maximum) number of memories, $N$, generally helps the performance although there could be some variance, as in Table 9c. We use a default value of 6 past frames to strike a balance between temporal context length and computational cost. Using fewer channels for memories does not cause much performance regression as in Table 9d, while making the memory required for storage $4\times$ smaller.

| | $\mathcal{J}\&\mathcal{F}$ | | | | mIoU |
|---|---|---|---|---|---|
| res. | MOSE dev | SA-V val | 9 zero-shot | speed | SA-23 |
| 512 | 73.0 | 68.3 | 70.7 | **1.00×** | 59.7 |
| 768 | 76.1 | **71.1** | **72.5** | 0.43× | 61.0 |
| 1024 | **77.0** | 70.1 | 72.3 | 0.22× | **61.5** |

(a) **Resolution.**

| | $\mathcal{J}\&\mathcal{F}$ | | | | mIoU |
|---|---|---|---|---|---|
| #frames | MOSE dev | SA-V val | 9 zero-shot | speed | SA-23 |
| 4 | 71.1 | 60.0 | 67.7 | 1.00× | **60.1** |
| 8 | 73.0 | **68.3** | 70.7 | 1.00× | 59.7 |
| 10 | **74.5** | 68.1 | **71.1** | 1.00× | 59.9 |

(b) **#Frames.**

| | $\mathcal{J}\&\mathcal{F}$ | | | | mIoU |
|---|---|---|---|---|---|
| #mem. | MOSE dev | SA-V val | 9 zero-shot | speed | SA-23 |
| 4 | **73.5** | 68.6 | 70.5 | **1.01×** | 59.9 |
| 6 | 73.0 | 68.3 | **70.7** | 1.00× | 59.7 |
| 8 | 73.2 | **69.0** | 70.7 | 0.93× | 59.9 |

(c) **#Memories.**

| | $\mathcal{J}\&\mathcal{F}$ | | | | mIoU |
|---|---|---|---|---|---|
| chan. dim. | MOSE dev | SA-V val | 9 zero-shot | speed | SA-23 |
| 64 | 73.0 | **68.3** | **70.7** | **1.00×** | 59.7 |
| 256 | **73.4** | 66.4 | 70.0 | 0.92× | **60.0** |

(d) **Memory channels.**

| | $\mathcal{J}\&\mathcal{F}$ | | | | mIoU |
|---|---|---|---|---|---|
| (#sa, #ca) | MOSE dev | SA-V val | 9 zero-shot | speed | SA-23 |
| (2, 2) | **73.3** | 67.3 | 70.2 | **1.13×** | 59.9 |
| (3, 2) | 72.7 | 64.1 | 69.5 | 1.08× | **60.0** |
| (4, 4) | 73.0 | **68.3** | **70.7** | 1.00× | 59.7 |

(e) **Memory attention.**

| | $\mathcal{J}\&\mathcal{F}$ | | | | mIoU |
|---|---|---|---|---|---|
| img. enc. | MOSE dev | SA-V val | 9 zero-shot | speed | SA-23 |
| S | 70.9 | 65.5 | 69.4 | **1.33×** | 57.8 |
| B+ | 73.0 | **68.3** | 70.7 | 1.00× | 59.7 |
| L | **75.0** | 66.3 | **71.9** | 0.60× | **61.1** |

(f) **Image encoder size**.

Table 9: We ablate modeling capacity along input size (resolution, #frames), memory size (#memories, memory channel dim) and model size (memory attention, image encoder). Ablation defaults in  gray .

**Model size.** More capacity in the image encoder or memory-attention (#self-/#cross-attention blocks) generally leads to improved results, as shown in Tables 9e, 9f. Scaling the image encoder brings gains on both image and video metrics, while scaling the memory-attention only improves video metrics. We default to using a B+ image encoder, which provides a reasonable balance for speed and accuracy.

### A.2.2 RELATIVE POSITIONAL ENCODING

| RPB | 2d-RoPE | $\mathcal{J}\&\mathcal{F}$ | | | | | mIoU |
|---|---|---|---|---|---|---|---|
| | | MOSE dev | SA-V val | LVOSv2 val | 9 zero-shot | speed | SA-23 |
| | ✓ | 73.0 | **68.3** | **71.6** | 70.7 | 1.00× | 59.7 |
| ✓ | ✓ | **73.6** | 67.9 | 71.0 | **71.5** | 0.93× | **60.0** |
| | | 72.8 | 67.1 | 70.3 | 70.3 | **1.04×** | 59.9 |

Table 10: We use 2d-RoPE positional encoding in memory attention while removing RPB from the image encoder by default ( gray ). Removing RPB also allows us to enable FlashAttention-2 (Dao, 2023), which gives a significant speed boost at ($1024^2$) resolution. At such higher resolution , the speed gap between 2d-RoPE (1st row) and the no RoPE baseline (3rd row) becomes small (4%).

By default, we always use absolute positional encoding in both the image encoder as well as memory attention. In Table 10, we study relative positional encoding design choices. Here we also evaluate on LVOSv2 (Hong et al., 2024) with 3 clicks on the 1st frame as a benchmark for long-term video object segmentation.

While SAM (Kirillov et al., 2023) follows Li et al. (2022c) in adding relative positional biases (RPB) to all image encoder layers, Bolya et al. (2023) improve upon this by removing RPB in all but the global attention layers while adopting "absolute-win" positional encoding which brings large speed gains. We improve upon this further by removing all RPB from the image encoder, with no performance regression on SA-23 and minimal regression on video benchmarks (see Table 10), while giving a significant speed boost at 1024 resolution. We also find it is beneficial to use 2d-RoPE (Su et al., 2021; Heo et al., 2024) in the memory attention.

### A.2.3 MEMORY ARCHITECTURE ABLATIONS

**Recurrent memory.** We investigate the effectiveness of feeding the memory features to a GRU before adding them to the memory bank. Similar to §A.2.2, we also evaluate on LVOSv2 as an additional

benchmark for long-term object segmentation. While prior works have commonly employed GRU (Cho et al., 2014) states as a means of incorporating memory into the tracking process, our findings in Table 11 suggest that this approach does not provide an improvement (except slightly on LVOSv2). Instead, we find it sufficient to directly store the memory features in the memory bank, which is both simpler and more efficient.

**Object pointers.** We ablate the impact of cross-attending to the object pointer vectors from the mask decoder output in other frames (see §4). The results presented in Table 11 show that while cross-attending to object pointers does not enhance average performance across the 9 zero-shot datasets, it significantly boosts performance on SA-V val dataset as well as on the challenging LVOSv2 benchmark (validation split). Hence, we default to cross-attending to object pointers together with the memory bank embeddings from the memory encoder.

| Object Pointers | GRU | $\mathcal{J}\&\mathcal{F}$ | | | | speed | mIoU |
| | | MOSE dev | SA-V val | LVOSv2 val | 9 zero-shot | | SA-23 |
|---|---|---|---|---|---|---|---|
| | | **73.1** | 64.5 | 67.0 | **70.9** | **1.00×** | 59.9 |
| | ✓ | 72.3 | 65.3 | 68.9 | 70.5 | 0.97× | **60.0** |
| ✓ | | 73.0 | **68.3** | **71.6** | 70.7 | **1.00×** | 59.7 |

Table 11: Ablations on memory design. We use object pointers by default ( gray ) and also study recurrent GRU memory.

# B    DETAILS ON THE PVS TASK

The Promptable Visual Segmentation (PVS) task can be seen as an extension of the Segment Anything (SA) task from static images to videos. In the PVS setting, given an input video, the model can be interactively *prompted* with different types of inputs (including clicks, boxes, or masks) on any frame in the video, with the goal of segmenting (and tracking) a valid object throughout the video. When interacting with a video, the model provides an instant response on the frame being prompted (similar to the interactive experience of SAM on images), and also returns the segmentation of the object throughout the entire video in near real-time. Similar to SAM the focus is on *valid* objects which have a clearly defined boundary, and we do *not* consider regions without visual boundaries (e.g. Bekuzarov et al. (2023)). Fig. 7 illustrates the task.

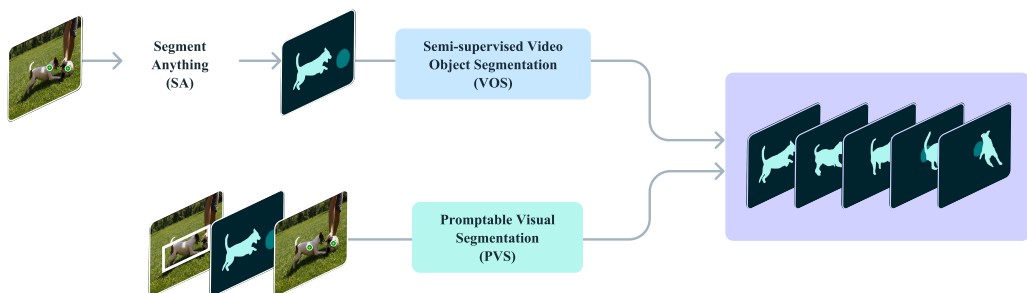

Figure 7: An illustration of the Promptable Visual Segmentation task (PVS). Previously studied tasks such as Segment Anything (SA) and semi-supervised Video Object Segmentation (VOS) can be seen as special cases of the PVS task.

PVS is related to tasks in the image and video domains. For images, the SA task can be considered a subset of PVS with the video reduced to a single frame. Similarly, traditional semi-supervised and interactive VOS (Pont-Tuset et al., 2017) tasks are special cases of PVS, limited to mask prompts provided only on the *first* frame and *scribbles* on multiple frames to segment objects throughout a video, respectively. In PVS, prompts can either be clicks, masks, or boxes, and the focus is on enhancing the interactive experience, enabling refinement of a segmentation with minimal interaction.

## C    LIMITATIONS

SAM 2 demonstrates strong performance in both static image and video domains, yet it encounters difficulties in certain scenarios. The model may fail to segment objects across shot changes and can lose track of or confuse objects in crowded scenes, after long occlusions or in extended videos. To alleviate this issue, we designed the ability to prompt SAM 2 in *any* frame: if the model loses the object or makes an error, refinement clicks on additional frames can quickly recover the correct prediction in most cases. SAM 2 also struggles with accurately tracking objects with very thin or fine details especially when they are fast-moving. Another challenging scenario occurs when there are nearby objects with similar appearance (e.g., multiple identical juggling balls). Incorporating more explicit motion modeling into SAM 2 could mitigate errors in such cases.

While SAM 2 can track multiple objects in a video simultaneously, SAM 2 processes each object separately, utilizing only shared per-frame embeddings without inter-object communication. While this approach is simple, incorporating shared object-level contextual information could aid in improving efficiency.

Our data engine relies on human annotators to verify masklet quality and select frames that require correction. Future developments could include automating this process to enhance efficiency.

SAM 2 tackles the promptable visual segmentation task, and the capabilities of SAM 2 do not extend to semantic recognition tasks. We believe that the advances made by SAM 2 can serve as a foundation for future work in recognition for both images and videos, similar to how SAM or SA-1B dataset was extended in (Li et al., 2023a; Yuan et al., 2024).

## D    SAM 2 DETAILS

### D.1    ARCHITECTURE

Here we discuss further architecture details, expanding on the model description in §4.

**Image encoder.** We use a feature pyramid network (Lin et al., 2017) to fuse the stride 16 and 32 features from Stages 3 and 4 of the Hiera image encoder respectively to produce the image embeddings for each frame. In addition, the stride 4 and 8 features from Stages 1 and 2 are not used in the memory attention but are added to the upsampling layers in the mask decoder as shown in Figure 8, which helps produce high-resolution segmentation details. We follow Bolya et al. (2023) in using *windowed* absolute positional embeddings in the Hiera image encoder. In Bolya et al. (2023), RPB provided positional information spanning *across* windows in the image encoder, in lieu of which we adopt a simpler approach of interpolating the global positional embedding instead to span across windows. We do not use any relative positional encoding. We train models with varying image encoder sizes – T, S, B+ and L. We follow Li et al. (2022c) and use global attention in only a subset of the image encoder layers (see Table 12).

**Memory attention.** In addition to sinusoidal absolute positional embeddings, we use 2d spatial Rotary Positional Embedding (RoPE) (Su et al., 2021; Heo et al., 2024) in self-attention and cross-attention layers. The object pointer tokens are excluded from RoPE as they do not have specific spatial correspondence. By default, the memory attention uses $L = 4$ layers.

**Prompt encoder and mask decoder.** The prompt encoder design follows SAM, and we next discuss additional details on design changes in the mask decoder. We use the mask token corresponding to the output mask as the object pointer token for the frame, which is placed in the memory bank. As discussed in §4, we also introduce an occlusion prediction head. This is accomplished by including an additional token along with the mask and IoU output tokens. An additional MLP head is applied to this new token to produce a score indicating the likelihood of the object of interest being visible in the current frame (as shown in Figure 8). In the memory bank, we also add a learned occlusion embedding to the memory features of those frames that are predicted to be occluded (invisible) by the occlusion prediction head.

SAM introduced the ability to output multiple valid masks when faced with ambiguity about the object being segmented in an image. For example, when a person clicks on the tire of a bike, the

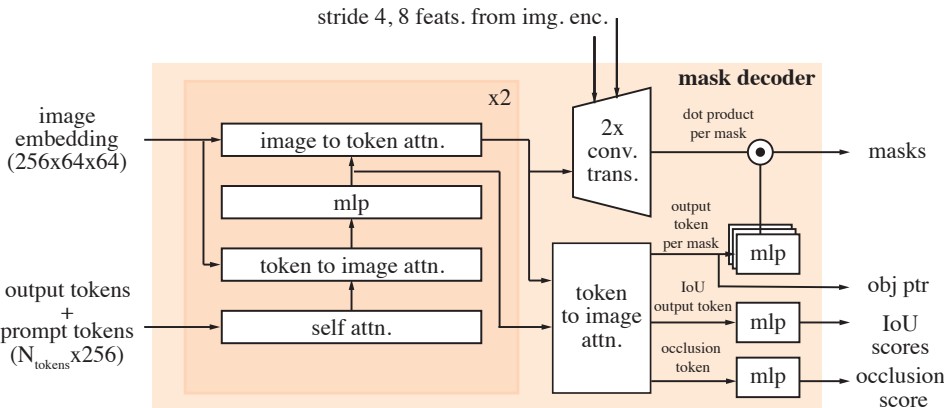

Figure 8: Mask decoder architecture. The design largely follows SAM, and we additionally include the stride 4 and 8 features from the image encoder during upsampling. We also use the mask token corresponding to the output mask as an object pointer and generate an occlusion score which indicates if the object of interest is visible in the current frame.

model can interpret this click as referring to only the tire or the entire bike and output multiple predictions. In videos, this ambiguity can extend across video frames. For example, if in one frame only the tire is visible, a click on the tire might relate to just the tire, or as more of the bike becomes visible in subsequent frames, this click could have been intended for the entire bike. To handle this ambiguity, SAM 2 predicts multiple masks at each step of the video. If further prompts do not resolve the ambiguity, the model selects the mask with the highest predicted IoU for the current frame for further propagation in the video.

**Memory encoder and memory bank.** Our memory encoder does not use an additional image encoder and instead reuses the image embeddings produced by the Hiera encoder, which are fused with the predicted mask information to produce memory features (as discussed in §4). This design allows the memory features to benefit from the strong representations produced by the image encoder (especially when we scale the image encoder to a larger size). Further, we project the memory features in our memory bank to a dimension of 64, and split the 256-dim object pointer into 4 tokens of 64-dim for cross-attention to the memory bank.

**Handling multiple objects in a video.** When applying SAM 2 to segment multiple objects in the same video (such as multi-object tracking in the semi-supervised VOS evaluation), we perform inference on each object independently. More specifically, we share the visual features from the image encoder between all the objects in the video but run all the other model components (such as the memory bank and the mask decoder) separately for each object.

## D.2 TRAINING

### D.2.1 PRE-TRAINING

We first pre-train SAM 2 on static images on the SA-1B dataset. Table 12a details the settings used during pre-training on SA-1B – other settings not mentioned here follow Kirillov et al. (2023). The image encoder is initialized from MAE pre-trained Hiera (Ryali et al., 2023). Similar to SAM, we filter masks covering more than 90% of the image and restricted training to 64 randomly sampled masks per image.

Unlike SAM, we found it beneficial to use an $\ell_1$ loss to more aggressively supervise the IoU predictions and to apply a sigmoid activation to the IoU logits to restrict the output into the range between 0 and 1. For multi-mask predictions (on the first click), we supervise the IoU predictions of *all* masks to encourage better learning of when a mask might be bad, but only supervise the mask logits with the lowest segmentation loss (linear combination of focal and dice loss). In SAM, during iterative sampling of points, two iterations were inserted with no additional prompts (only feeding the previous mask logits) – we do not add such iterations during our training and use 7 correction

clicks (instead of 8 in SAM). We also employ horizontal flip augmentation during training and resize the image to a square size of 1024×1024.

We use AdamW (Loshchilov & Hutter, 2019) and apply layer decay (Clark et al., 2020) on the image encoder and follow a reciprocal square-root schedule (Zhai et al., 2022). See Table 12 (a) for the hyperparameters in our pre-training stage.

### D.2.2  FULL TRAINING

After pre-training, we train SAM 2 on our introduced datasets SA-V + Internal (section §5.2), a 10% subset of SA-1B, and a mixture of open-source video datasets including DAVIS (Pont-Tuset et al., 2017; Caelles et al., 2019), MOSE (Ding et al., 2023), and YouTubeVOS (Xu et al., 2018b). Our released model is trained on SA-V manual + Internal and SA-1B.

SAM 2 is designed for two tasks; the PVS task (on videos) and the SA task (on images). Training is done *jointly* on image and video data. To optimize our data usage and computational resources during training, we adopt an alternating training strategy between video data (multiple frames) and static images (one single frame). Specifically, in each training iteration, we sample a full batch either from the image or video dataset, with their sampling probabilities proportional to the size of each data source. This approach allows for a balanced exposure to both tasks and a different batch size for each data source to maximize compute utilization. Settings not explicitly mentioned here for the image task follow settings from the pre-training phase. See Table 12 (b) for the hyperparameters in our full training stage. The training data mixture consists of ∼15.2% SA-1B, ∼70% SA-V and ∼14.8% Internal. The same settings are used when open-source datasets are included, with the change that the additional data is included (∼1.3% DAVIS, ∼9.4% MOSE, ∼9.2% YouTubeVOS, ∼15.5% SA-1B, ∼49.5% SA-V, ∼15.1% Internal). When training on SA-V and other video datasets, we only use those manually annotated masklets (without adding automatically generated ones), which are sufficient to achieve strong performance based on our analyses.

We apply a series of data augmentations to the training videos (detailed in Table 12), including random horizontal flips, random affine transforms, random color jittering, and random grayscale transforms, as listed in Table 12. We also adopt a mosaic transform to simulate challenging scenarios with multiple similar-looking objects – with 10% probability, we tile the same training video into a 2×2 grid and select a masklet from one of the 4 quadrants as the target object to segment. In this case, the model must focus on other cues like motion or temporal continuity to distinguish the target object from their identical-looking counterparts in other quadrants. In addition, the videos and objects in each quadrant are smaller in size (only half the original width and height) after this mosaic transform, which also facilitates learning to segment small objects.

We train by simulating an interactive setting, sampling 8-frame sequences and randomly selecting up to 2 frames (including the first) for corrective clicks. During training, we use ground-truth masklets and model predictions to sample prompts, with initial prompts being the ground-truth mask (50% probability), a positive click from the ground-truth mask (25%), or a bounding box input (25%).

We restrict the maximum number of masklets for each sequence of 8 frames to 3 randomly chosen ones. We reverse the temporal order with a probability of 50% to help generalization to bi-directional propagation. When we sample corrective clicks, with a small probability of 10%, we randomly sample clicks from the ground truth mask, irrespective of the model prediction, to allow additional flexibility in mask refinement.

**Fine-tuning using 16-frame sequences.** A potential shortcoming of the procedure above is that the model only sees sampled 8-frame sequences during training, which is relatively short compared to the full video length during inference. To alleviate this issue and further boost the segmentation quality on long videos, we introduce an extra fine-tuning stage where we sample 16-frame sequences on challenging videos (those videos with the highest number of edited frames, as described in §E.2.1). More specifically, we sort our masklets by number of edited frames and only consider the top 50% most edited masklets for training, for both SA-V and Internal datasets. We still keep the complete versions of the OSS datasets (DAVIS, MOSE, and YouTubeVOS) in the training mix. We fine-tune for 50k iterations (1/3 of the original schedule) using half of the original learning rate and freeze the image encoder to fit the 16-frame sequence into the 80 GB memory of A100 GPUs.

| config | value |
|---|---|
| data | SA-1B |
| steps | $\sim$90k |
| resolution | 1024 |
| precision | bfloat16 |
| optimizer | AdamW |
| optimizer momentum | $\beta_1, \beta_2{=}0.9, 0.999$ |
| gradient clipping | type: $\ell_2$, max: 0.1 |
| weight decay | 0.1 |
| learning rate (lr) | 4e-4 |
| lr schedule | reciprocal sqrt, timescale=1000 |
| warmup | linear, 1k iters |
| cooldown | linear, 5k iters |
| layer-wise decay | 0.8 (T, S), 0.9 (B+), 0.925 (L) |
| augmentation | hflip, resize to 1024 (square) |
| batch size | 256 |
| drop path | 0.1 (T, S), 0.2 (B+), 0.3 (L) |
| mask losses (weight) | focal (20), dice (1) |
| IoU loss (weight) | $\ell_1$ (1) |
| max # masks per image | 64 |
| # correction points | 7 |
| global attn. blocks | 5-7-9 (T), 7-10-13 (S), 12-16-20 (B+), 23-33-43 (L) |

(a) Pre-training

| config | value |
|---|---|
| data | SA-1B, Internal, SA-V |
| steps | $\sim$150k |
| resolution | 1024 |
| precision | bfloat16 |
| optimizer | AdamW |
| optimizer momentum | $\beta_1, \beta_2{=}0.9, 0.999$ |
| gradient clipping | type: $\ell_2$, max: 0.1 |
| weight decay | 0.1 |
| learning rate (lr) | img. enc.: 6e-5, other: 3.0e-4 |
| lr schedule | cosine |
| warmup | linear, 7.5k iters |
| layer-wise decay | 0.8 (T, S), 0.9 (B+), 0.925 (L) |
| image augmentation | hflip, resize to 1024 (square) |
| video augmentation | hflip, affine (deg: 25, shear: 20), colorjitter (b: 0.1, c: 0.03, s: 0.03, h: null), grayscale (0.05), per frame colorjitter (b: 0.1, c: 0.05, s: 0.05, h: null), mosaic-2$\times$2 (0.1) |
| batch size | 256 |
| drop path | 0.1 (T, S), 0.2 (B+), 0.3 (L) |
| mask losses (weight) | focal (20), dice (1) |
| IoU loss (weight) | $\ell_1$ (1) |
| occlusion loss (weight) | cross-entropy (1) |
| max. masks per frame. | image: 64, video: 3 |
| # correction points | 7 |
| global attn. blocks | 5-7-9 (T), 7-10-13 (S), 12-16-20 (B+), 23-33-43 (L) |

(b) Full training

Table 12: Hyperparameters and details of SAM 2 pre-training and full training. Note that some settings vary with image encoder size (T, S, B+, L).

**Losses and optimization.** We supervise the model's predictions using a linear combination of focal and dice losses for the mask prediction, mean-absolute-error (MAE) loss for the IoU prediction, and cross-entropy loss for object prediction with a ratio of 20:1:1:1 respectively.

As during pre-training, for multi-mask predictions, we only supervise the mask with the lowest segmentation loss. If the ground-truth does not contain a mask for a frame, we do not supervise any of the mask outputs (but always supervise the occlusion prediction head that predicts whether there should exist a mask in the frame).

### D.3 SPEED BENCHMARKING

We conduct all benchmarking experiments on a single A100 GPU using PyTorch 2.3.1 and CUDA 12.1, under automatic mixed precision with bfloat16. We compile the image encoder with `torch.compile` for all SAM 2 models and do the same for SAM and HQ-SAM for direct comparison on the SA task (Tables 5 and 16). The FPS measurements for the SA task were conducted using a batch size of 10 images, which was found to yield the highest FPS across all three model types. For video tasks, we use a batch size of 1 following the common protocol in video segmentation.

## E DATA DETAILS

### E.1 SA-V DATASET DETAILS

**Videos.** Resolutions range from 240p to 4K with $1{,}401 \times 1{,}037$ on average. Duration ranges from 4 seconds to 2.3 minutes, with an average of 13.8 seconds, totaling 4.2M frames and 196 hours.

**Dataset diversity.** As shown in Fig. 10, SA-V videos were recorded across 47 countries (Fig. 10b), by diverse participants (self-reported demographics in Fig. 10c). Fig. 10a shows a comparison of mask size distribution (normalized by video resolution) with DAVIS, MOSE, and YouTubeVOS. More than $88\%$ of SA-V masks have a normalized mask area less than 0.1.

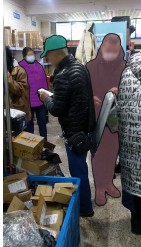 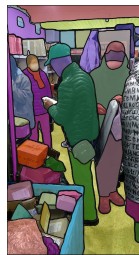

| (a) ML only | (b) With Auto |

Figure 9: Annotations overlaid on the first frame: (a) manual labels (ML) only, (b) with automatic labels (Auto). Automatic labels increase diversity and coverage.

**Automatic masklets.** Similar to the approach in Kirillov et al. (2023), automatic masklets are generated by prompting the model with regular grids. We prompt the model with a $32 \times 32$ grid on the first frame, and we use a $16 \times 16$ grid on 4 zoomed image crops of the first frame (derived from a $2 \times 2$ overlapped window) and a $4 \times 4$ grid on 16 zoomed image crops of the first frame (derived from a $4 \times 4$ overlapped window). We apply two post-processing steps across all frames. First, we remove tiny disconnected components with areas smaller than 200 pixels. Second, we fill in holes in segmentation masks if the area of the hole is less than 200 pixels. By combining automatically generated with manual masklets, we enhance the coverage of annotations in the SA-V dataset, see Fig. 9.

### E.1.1 FAIRNESS EVALUATION

We evaluate SAM 2 for fairness across demographic groups. We collect annotations for the people category in the Ego-Exo4D (Grauman et al., 2023) dataset, which contains self-reported demographic information supplied by the subject of the video. We employ the same annotation setup as for SA-V val and test sets and apply this to 20-second clips from the third-person (exo) videos. We evaluate SAM 2 on this data using 1-, 3-clicks, and ground-truth mask on the first frame.

| | 1-click | 3-click | mask |
|---|---|---|---|
| *gender* | | | |
| male | 81.9 | 95.1 | 95.9 |
| female | 75.1 | 94.1 | 95.2 |
| *age* | | | |
| 18-26 | 77.2 | 95.0 | 95.7 |
| 26-50 | 76.7 | 94.7 | 95.8 |
| 50+ | 81.4 | 95.1 | 96.2 |

Table 13: Fairness evaluation of SAM 2 (under $\mathcal{J}\&\mathcal{F}$ metric) on protected demographic groups.

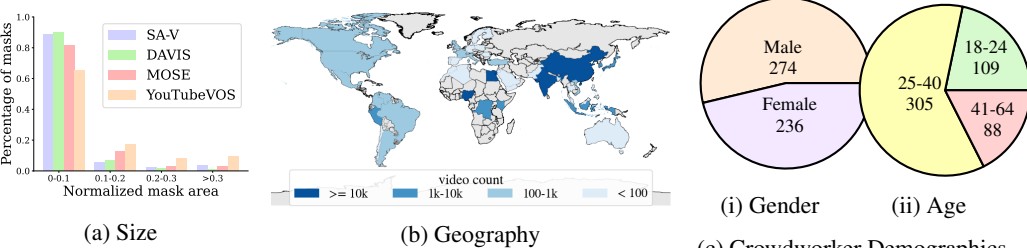

| (a) Size | (b) Geography | (c) Crowdworker Demographics |

Figure 10: Dataset distribution: (a) masklets size distribution (normalized by video resolution), (b) geographic diversity of the videos, and (c) self-reported demographics of the crowdworkers who recorded the videos.

Table 13 shows the comparison in $\mathcal{J}\&\mathcal{F}$ accuracy of SAM 2 for segmenting people across gender and age. At 3 clicks and with ground-truth mask prompts there is minimal discrepancy. We manually inspect 1 click predictions, and find the model frequently predicts the mask for a part instead of the person. When limiting the comparison to clips where the person is correctly segmented, the gap in 1 click shrinks substantially ($\mathcal{J}\&\mathcal{F}$ male 94.3, female 92.7), suggesting the discrepancy can be partially attributed to ambiguity in the prompt.

In Appendix J, we provide model, data and annotation cards for SA-V.

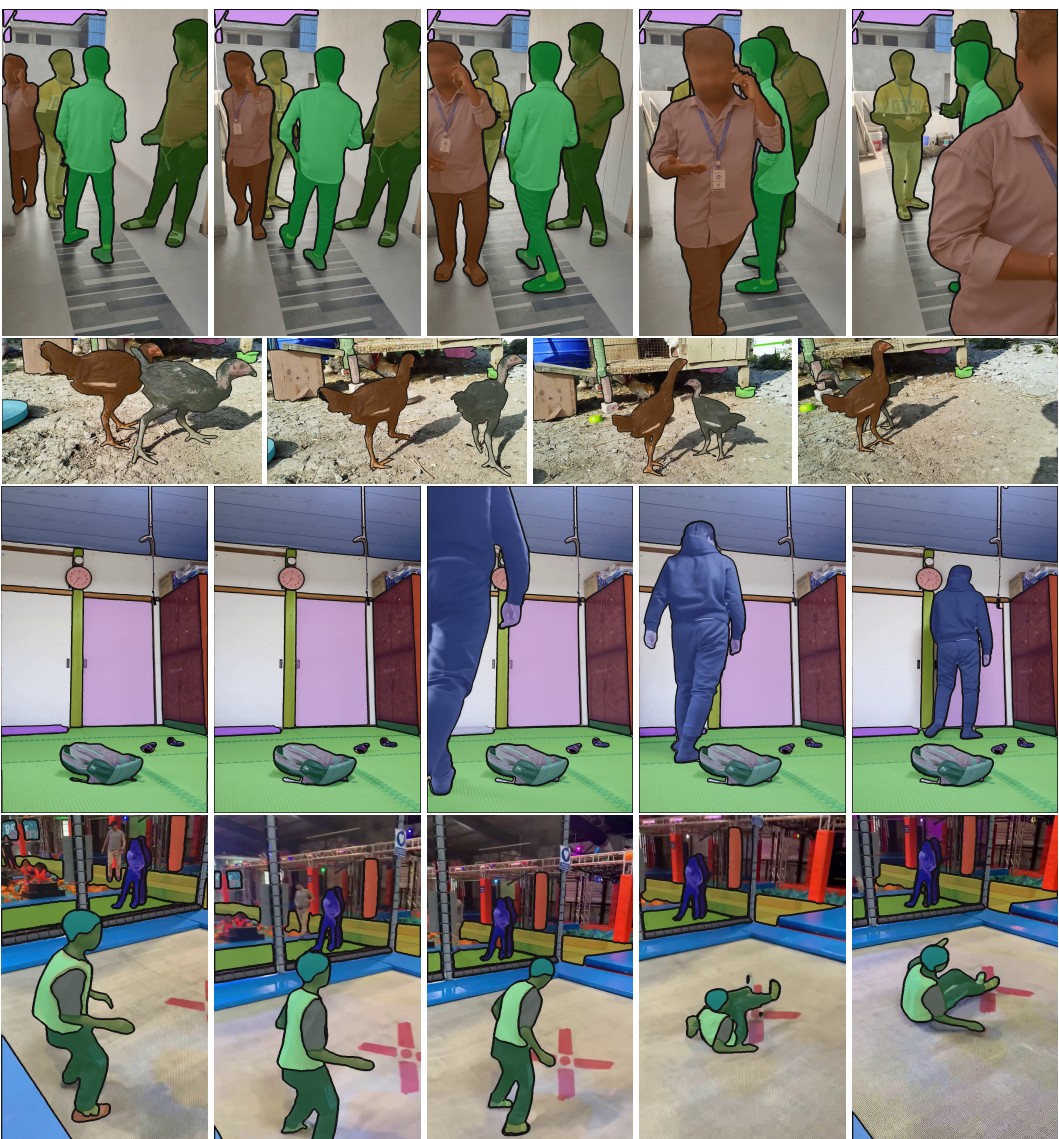

Figure 11: Example videos from the SA-V dataset with masklets overlaid (manual and automatic). Each masklet has a unique color, and each row represents frames from one video, with 1 second between them.

### E.2 DATA ENGINE DETAILS

#### E.2.1 ANNOTATION PROTOCOL

A diagram of the annotation protocol used in our data engine is shown in Fig. 12. The annotation task was separated into steps each carried out by a different annotator: Steps 1 and 2 focus on *object*

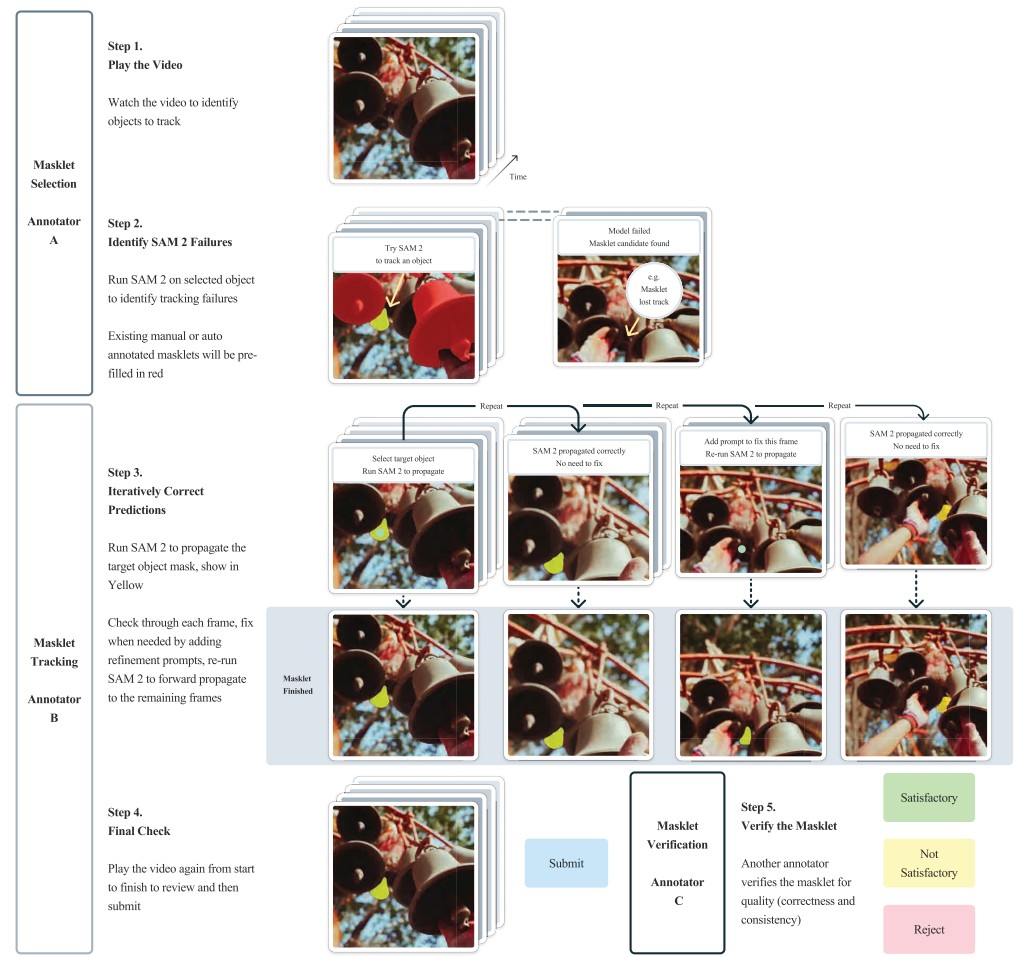

Figure 12: Annotation guideline overview. There are 3 main annotation tasks: masklet selection, masklet tracking, and masklet verification. Each task has a different set of annotators working on it.

*selection*, Steps 3 and 4 on *masklet tracking*, and Step 5 on *quality verification*. SAM 2 was deployed on GPU as an API and built into the annotation tool to enable interactive use.

Compared to image segmentation annotation, large-scale video segmentation annotation presents unique challenges which require innovations in the annotation task design and protocol. To improve our model's ability to "segment anything", it was important to focus annotation on challenging objects where SAM 2 struggled. We leveraged our online model in the loop setup to enable this, requesting annotators to use SAM 2 interactively to identify failure modes and then correct them.

We found the number of edited frames to be a proxy to the "challengingness" of an object as shown in Table 8. Therefore, we asked annotators to annotate objects that required at least 2 edited frames with SAM 2 in the loop. To focus annotation on less prominent and more challenging cases, annotators were presented with videos pre-filled with verified satisfactory *automatic* masklets and asked to find un-annotated challenging objects. We further decouple the object selection task from the annotation task: in the selection task annotators focus on choosing the challenging objects in one frame, while in the annotation task annotators are presented with a challenging target object and requested to annotate the masklet consistently throughout the video.

### E.2.2 DATA ENGINE PHASE COMPARISON

The comparison of data engine phases shown in Table 1 was conducted as a controlled experiment using 169 videos and 452 masklets. We ask three subsets of annotators to annotate the same set of objects with the annotation protocol from each phase. We categorize masklets into three buckets based on the mask area in the first frame (small: 1 to $32^2$, medium: $32^2$ to $96^2$, and large: equal or greater than $96^2$). Phase 1 data is used as the quality reference, due to the high quality masks from frame-by-frame manual annotation with SAM.

## F DETAILS ON ZERO-SHOT TRANSFER EXPERIMENTS

In this section, we describe further details of our zero-shot experiments (§6). Unless otherwise noted, the results reported in this section follow our default setup using Hiera-B+ image encoder with a resolution of 1024 and trained on the full combination of datasets, i.e., SAM 2 (Hiera-B+) in Table 6.

### F.1 ZERO-SHOT VIDEO TASKS

#### F.1.1 VIDEO DATASET DETAILS

We evaluate SAM 2 on a diverse benchmark of 17 zero-shot datasets: EndoVis 2018 (Allan et al., 2020) contains medical surgery videos with robotic instruments. ESD (Huang et al., 2023) contains videos from a robot manipulator camera often with motion blur. LVOSv2 (Hong et al., 2024) is a benchmark for long-term video object segmentation. LV-VIS (Wang et al., 2023) contains videos from a diverse set of open-vocabulary object categories. UVO (Wang et al., 2021b) contains videos for open-world object segmentation, and VOST (Tokmakov et al., 2022) contains videos with objects undergoing large transformations such as egg broken or paper torn. PUMaVOS (Bekuzarov et al., 2023) contains videos with segments around object parts such as a person's cheek. Virtual KITTI 2 (Cabon et al., 2020) is a synthetic video dataset with driving scenes. VIPSeg (Miao et al., 2022) provides object segmentation in panoptic videos. Wildfires (Toulouse et al., 2017) contains wildfire videos under different conditions from the Corsican Fire Database. VISOR (Darkhalil et al., 2022) contains egocentric videos in kitchen scenes with segments around hands and active objects. FBMS (Brox et al., 2010) provides motion segmentation over moving objects in videos. Ego-Exo4D (Grauman et al., 2023) is a large dataset with egocentric videos around various human activities. Cityscapes (Cordts et al., 2016) contains videos for urban driving scenes. Lindenthal Camera (Haucke & Steinhage, 2021) contains videos in a wildlife park with segments around observed animals such as birds and mammals. HT1080WT Cells (Gómez-de Mariscal et al., 2021) contains microscopy videos with cell segments. Drosophila Heart (Fishman et al., 2023) contains microscopy videos for the heart of fruit flies.

Among these 17 zero-shot video datasets above, 9 of them (EndoVis, ESD, LVOSv2, LV-VIS, UVO, VOST, PUMaVOS, Virtual KITTI 2, and VIPSeg) have dense object segments annotated for every video frame. In the remaining 8 datasets (Wildfires, VISOR, FBMS, Ego-Exo4D, Cityscapes, Lindenthal Camera, HT1080WT Cells, and Drosophila Heart), the object segments are sparsely annotated over only a subset of video frames, and we compute the metrics on those frames where the ground-truth segmentation masks are available. In most evaluations of the paper, we only evaluate zero-shot performance on the 9 densely annotated datasets, while in our semi-supervised VOS evaluation (§6.2), we evaluate on all these 17 datasets listed above.

#### F.1.2 INTERACTIVE OFFLINE AND ONLINE EVALUATION DETAILS

*Offline evaluation* involves *multiple passes* over the entire video. We start with click prompts on the first frame, segment the object throughout the entire video, and then in the next pass, we select the frame with the lowest segmentation IoU w.r.t. the ground-truth as the new frame for prompting. The model then segments the object again throughout the video based on *all* prompts received previously, until reaching a maximum of $N_{\text{frame}}$ passes (with one new prompted frame in each pass).

*Online evaluation* involves only *one pass* over the entire video. We start with click prompts on the first frame and propagate the prompts across the video, pausing propagation when encountering a frame with a low-quality prediction (IoU $< 0.75$ with ground-truth). We then add additional click prompts on the paused frame to correct the segment on this frame and resume the propagation *forward*

until reaching another low quality frame with IoU $< 0.75$. This is repeated while the number of prompted frames is less than the maximum $N_{\text{frame}}$. Unlike the previous offline evaluation, in this setting, the new prompts only affect the frames *after* the current paused frame but not the frames before it.

In both settings, we evaluate on 9 densely annotated datasets in §F.1.1 (EndoVis, ESD, LVOSv2, LV-VIS, UVO, VOST, PUMaVOS, Virtual KITTI 2, and VIPSeg). If a video contains multiple objects to segment in its ground-truth annotations, we perform inference on each object independently. We simulate interactive video segmentation with $N_{\text{click}} = 3$ clicks per frame, assuming that the user would visually locate the object to label it (with initial clicks) or to refine the current segmentation prediction of it (with correction clicks). Specifically, when starting the first pass (where there are not any existing predictions yet), we place an initial click on the first frame at the center[1] of the object's ground-truth mask and then interactively add two more clicks based on the center of the error region (between the ground-truth mask and the predicted segments on the first frame). Then in subsequent passes (where there are already predicted segments), we interactively add three clicks based on the center of the error region (between the ground-truth mask and the predicted segments on the frame being prompted).

We report the average $\mathcal{J}\&\mathcal{F}$ metric over $N_{\text{frame}} = 1, \ldots, 8$ interacted frames and the $\mathcal{J}\&\mathcal{F}$ metrics under different annotation time on a video based on the following assumptions:

- On each frame, it takes $T_{\text{loc}} = 1$ sec for the annotator to visually locate an object in the frame, and $T_{\text{click}} = 1.5$ sec to add each click, following Delatolas et al. (2024).

- In *offline* mode, it takes $T_{\text{exam}} = 30$ sec on a 300-frame video to examine the results throughout the video in each round, including finding the frame with the worst segmentation quality to add corrections (and for longer or shorter videos, this time is proportional to the video length $L$, assuming the annotator could examine the results at 10 FPS).

- In *online* mode, it takes $T_{\text{exam}} = 30$ sec on a 300-frame video to follow the results throughout the video in total, including pausing at a frame with low quality for further corrections (and this time is proportional to the video length $L$ similar to the offline mode).

- The annotation time for an object is $(T_{\text{exam}} \cdot (L/300) + T_{\text{loc}} + T_{\text{click}} \cdot N_{\text{click}}) \cdot N_{\text{frame}}$ in offline mode and $T_{\text{exam}} \cdot (L/300) + (T_{\text{loc}} + T_{\text{click}} \cdot N_{\text{click}}) \cdot N_{\text{frame}}$ in online mode, where $L$ is the total frame number in the video, $N_{\text{frame}} = 1, \ldots, 8$ is the number of frames annotated (i.e., the number of interactive rounds), and $N_{\text{click}} = 3$ is the number of clicks per frame.[2]

We show per-dataset results of SAM 2 and the two baselines (SAM+XMem++ and SAM+Cutie, see their details below) for interactive offline and online evaluation in Fig. 13 and Fig. 14. SAM 2 outperforms both baselines with a notable margin on all datasets and settings.

### F.1.3 SEMI-SUPERVISED VOS EVALUATION DETAILS

In §6.2, we also compare with previous video tracking methods under the semi-supervised VOS setting (Pont-Tuset et al., 2017), where prompts (which can be foreground/background clicks, bounding boxes, or ground-truth object masks) are provided only on the first frame of the video. When using click prompts, we interactively sample either 1, 3 or 5 clicks on the first video frame, and then track the object based on these clicks. Following the click-based evaluation in prior work (Kirillov et al., 2023; Sofiiuk et al., 2022), the initial click is placed on the object center and subsequent clicks are obtained from the center of the error region.

Similar to the interactive setting, here we also use SAM+XMem++ and SAM+Cutie as two baselines. For click or box prompts, SAM is first used to handle click or bounding box inputs, and its output mask is then used as input to XMem++ or Cutie. For mask prompts, the ground-truth object masks on the first frame are directly used as input to XMem++ and Cutie – this is the standard semi-supervised VOS setting and evaluates XMem++ and Cutie without using SAM.

---

[1] The center of a mask is defined as the mask pixel that has the largest Euclidean distance to the mask boundary.

[2] We note that this estimation does not account for the model's tracking FPS. The intuition is that human annotators can only examine the results at a lower speed, and therefore the model's tracking time is covered by $T_{\text{exam}}$.

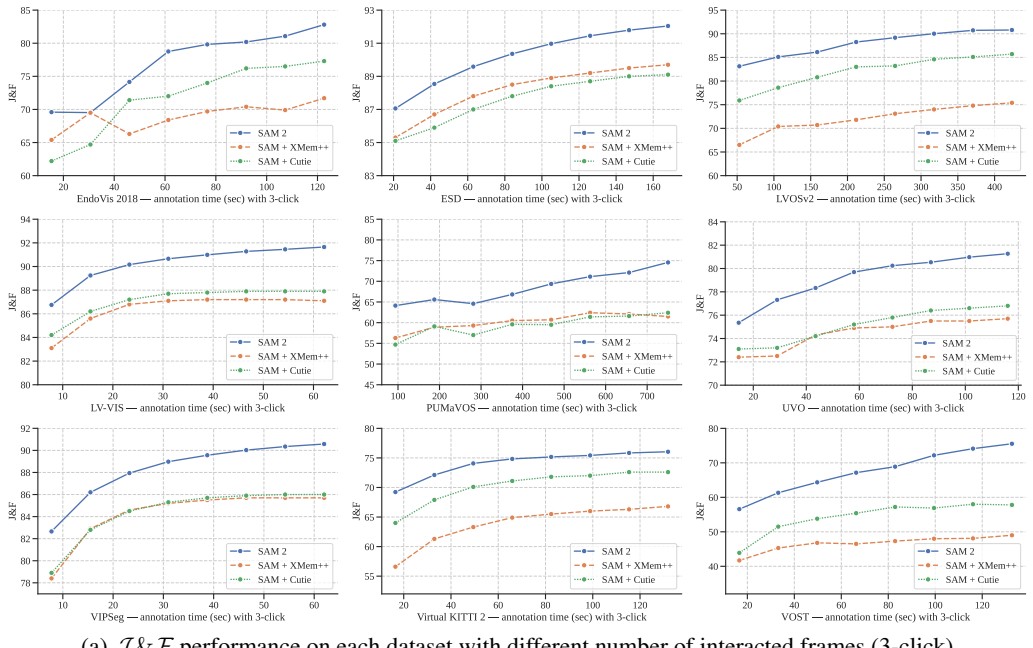

(a) $\mathcal{J}\&\mathcal{F}$ performance on each dataset with different number of interacted frames (3-click)

| Method | EndoVis 2018 | ESD | LVOSv2 | LV-VIS | PUMaVOS | UVO | VIPSeg | Virtual KITTI 2 | VOST | (average) |
|---|---|---|---|---|---|---|---|---|---|---|
| SAM + XMem++ | 68.9 | 88.2 | 72.1 | 86.4 | 60.2 | 74.5 | 84.2 | 63.8 | 46.6 | 71.7 |
| SAM + Cutie | 71.8 | 87.6 | 82.1 | 87.1 | 59.4 | 75.2 | 84.4 | 70.3 | 54.3 | 74.7 |
| SAM 2 | **77.0** | **90.2** | **87.9** | **90.3** | **68.5** | **79.2** | **88.3** | **74.1** | **67.5** | **80.3** |

(b) average $\mathcal{J}\&\mathcal{F}$ on each dataset over 8 interacted frames (3-click)

Figure 13: Zero-shot performance of SAM 2 vs baselines (SAM+XMem++ and SAM+Cutie) under interactive *offline* evaluation with different numbers of interacted frames, using 3 clicks per interacted frame. See §F.1.2 for details.

In this setting, we evaluate on all 17 zero-shot video datasets in §F.1.1. If a dataset does not follow the standard VOS format, we preprocess it into a format similar to MOSE (Ding et al., 2023). During processing, we ensure that all objects in each video have a valid non-empty segmentation mask on the first frame to be compatible with semi-supervised VOS evaluation. In case an object doesn't appear in the first frame, we create a separate video for it starting from the first frame where the object appears.

We report the standard $\mathcal{J}\&\mathcal{F}$ metric (Pont-Tuset et al., 2017) for this evaluation. If a dataset provides an official evaluation toolkit, we use it for evaluation (on the VOST dataset, we report the $\mathcal{J}$ metric instead, following its official protocol (Tokmakov et al., 2022)). The results are shown in Table 4, where SAM 2 performs better than both baselines on the majority of the 17 datasets across different types of prompts.

We show per-dataset results of SAM 2 and the two baselines (SAM+XMem++ and SAM+Cutie, see their details below) for semi-supervised VOS evaluation in Fig. 15. SAM 2 outperforms both baselines on the majority of these datasets across different types of prompts.

### F.1.4 SAM+XMEM++ AND SAM+CUTIE BASELINE DETAILS

We adopt SAM+XMem++ and SAM+Cutie as two baselines for promptable video segmentation, where the click (or box) prompts are first processed by SAM to obtain an object mask, and then XMem++ / Cutie models track this SAM mask across the video to obtain the final masklet. In these two baselines, SAM can be used to provide both an *initial* object mask on the first frame, or to *correct* an existing object mask output by XMem++ or Cutie. This is used for subsequent interacted frames during interactive offline and online evaluation, where new positive and negative clicks are provided as corrections over an existing mask.

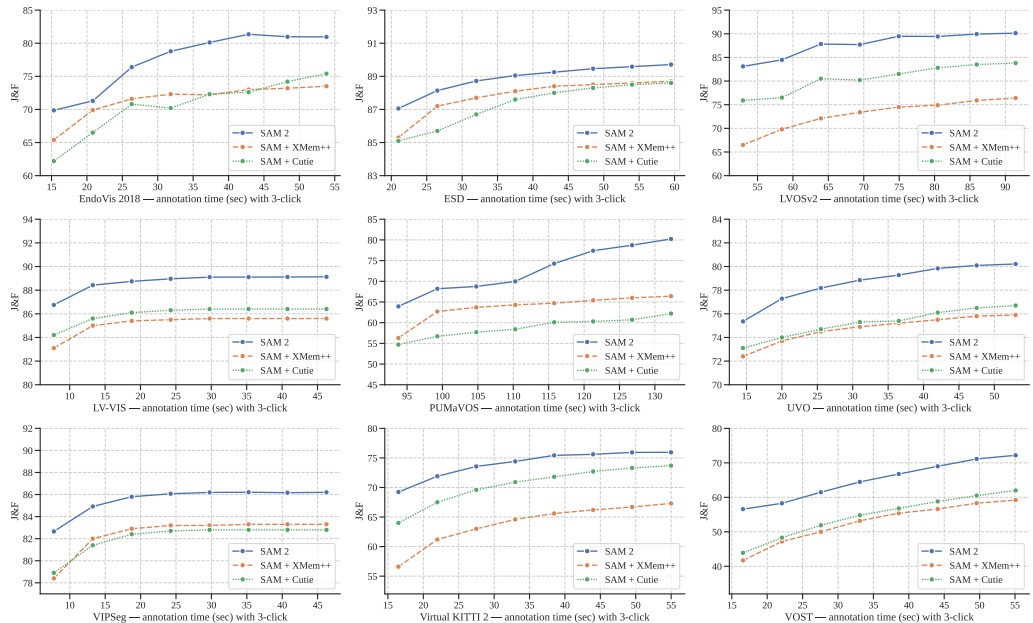

(a) $\mathcal{J}\&\mathcal{F}$ performance on each dataset with different number of interacted frames (3-click)

| Method | EndoVis 2018 | ESD | LVOSv2 | LV-VIS | PUMaVOS | UVO | VIPSeg | Virtual KITTI 2 | VOST | (average) |
|---|---|---|---|---|---|---|---|---|---|---|
| SAM + XMem++ | 71.4 | 87.8 | 72.9 | 85.2 | 63.7 | 74.7 | 82.5 | 63.9 | 52.7 | 72.8 |
| SAM + Cutie | 70.5 | 87.3 | 80.6 | 86.0 | 58.9 | 75.2 | 82.1 | 70.4 | 54.6 | 74.0 |
| SAM 2 | **77.5** | **88.9** | **87.8** | **88.7** | **72.7** | **78.6** | **85.5** | **74.0** | **65.0** | **79.8** |

(b) average $\mathcal{J}\&\mathcal{F}$ on each dataset over 8 interacted frames (3-click)

Figure 14: Zero-shot performance of SAM 2 vs baselines (SAM+XMem++ and SAM+Cutie) under interactive *online* evaluation with different numbers of interacted frames, using 3 clicks per interacted frame. See §F.1.2 for details.

When using SAM to apply a correction over an existing mask prediction in a given frame, we follow the strategy in EVA-VOS (Delatolas et al., 2024) to first initialize SAM with the XMem++ or Cutie output mask before incorporating the new correction clicks. Specifically, we first reconstruct the XMem++ or Cutie output mask in SAM by sampling clicks from them and feeding them as inputs to SAM until the reconstructed mask in SAM reaches IoU $> 0.8$ with the XMem++ or Cutie output mask. Then, to incorporate new positive and negative clicks for correction, we concatenate these additional correction clicks with the initial clicks sampled during mask construction, and feed the joint concatenated list as input into SAM to obtain the final corrected masks. We find that this strategy works better than several alternatives (such as feeding the XMem++ or Cutie output mask as a mask prompt together with new correction clicks into SAM, or taking only the correction clicks as inputs to SAM while ignoring the XMem++ or Cutie output mask).

## F.2 DAVIS INTERACTIVE BENCHMARK

We also evaluate on the DAVIS interactive benchmark (Caelles et al., 2018), which resembles our interactive offline evaluation in §6.1, where in each round of interaction, the evaluation server would provide new annotations on frames with the worst segmentation performance. The official DAVIS eval toolkit provides scribble prompts during interactions, while other work such as CiVOS (Vujasinović et al., 2022) has also extended this to cover click prompts.

Here we follow CiVOS to use positive and negative clicks as input prompts and adopt the same strategy for click sampling. We report the $\mathcal{J}\&\mathcal{F}$@60s and AUC-$\mathcal{J}\&\mathcal{F}$ metrics on this benchmark as provided by its evaluator, and compare to two baselines: MiVOS (Cheng et al., 2021b), which directly uses the provided scribbles via a scribble-to-mask module (and is also extended to click prompts in Vujasinović et al. (2022)), and CiVOS, which samples click from the provided scribbles.

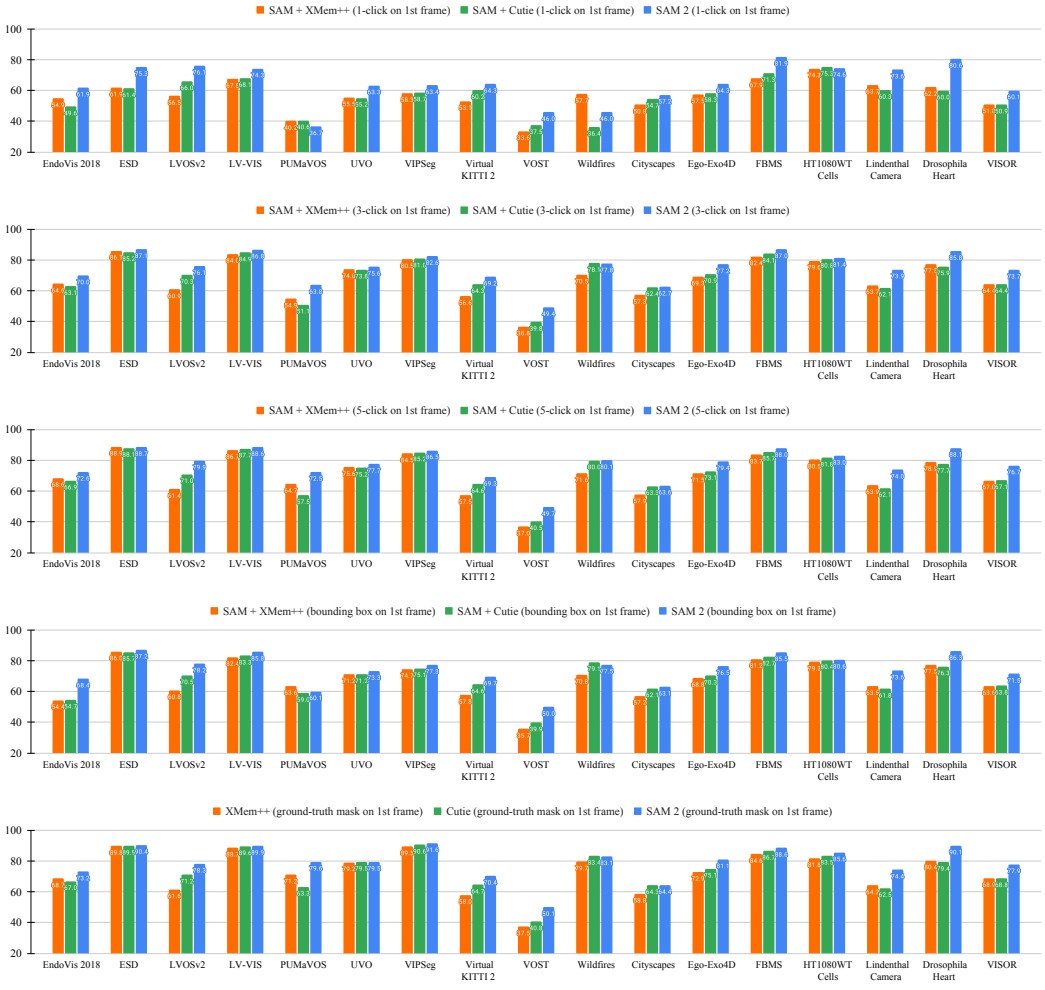

Figure 15: Zero-shot performance on 17 video datasets of SAM 2 vs two baselines (SAM+XMem++ and SAM+Cutie) under semi-supervised VOS evaluation using different prompts (1, 3 or 5 clicks, bounding boxes, or ground-truth masks on the first video frame), with the averaged performance across datasets for each type of prompt shown in Table 4 in the main text. See §F.1.3 for details.

| Method | input type | AUC-$\mathcal{J}\&\mathcal{F}$ | $\mathcal{J}\&\mathcal{F}$@60s |
|---|---|---|---|
| MA-Net (Miao et al., 2020) | scribbles | 0.79 | 0.79 |
| MiVOS (Cheng et al., 2021b) | scribbles | 0.87 | 0.88 |
| MiVOS (Cheng et al., 2021b)[‡] | clicks | 0.75 | 0.75 |
| CiVOS (Vujasinović et al., 2022) | clicks | 0.83 | 0.84 |
| **SAM 2** | clicks | **0.86** | **0.90** |

Table 14: Performance of SAM 2 and other models on the DAVIS interactive benchmark. For SAM 2, we use clicks as inputs following the click sampling strategy from CiVOS (Vujasinović et al., 2022). See §F.2 for details ([‡]: performance reported in Vujasinović et al. (2022)).

The results are shown in Table 14, where SAM 2 (based on click inputs) outperforms both baselines under click inputs. We note that SAM 2 often tends to segment object parts (e.g. a person's arm) on the first click while the DAVIS dataset mainly contains whole objects (e.g. an entire person), which could penalize SAM 2's $\mathcal{J}\&\mathcal{F}$ performance on this benchmark.

## F.3 VIPOSEG BENCHMARK

We further evaluate SAM 2 on the VIPOSeg benchmark (Xu et al., 2023), a large-scale benchmark for video object segmentation on panoptic wild scenes. We find that SAM 2 has strong performance on this benchmark, as shown in Table 15.

- In a *zero-shot* manner without training on this dataset, SAM 2 achieves $\mathcal{G} = 78.4$ on the VIPOSeg val split (where $\mathcal{G}$ is the overall metric defined in this benchmark, higher is better), which already outperforms the $\mathcal{G} = 78.2$ performance of the PAOT model (the best model in Table 2 of Xu et al. (2023) *trained* on this dataset).

- When *fine-tuned* on the VIPOSeg training split, SAM 2's performance is further improved to $\mathcal{G} = 79.7$ on the VIPOSeg val split.

- In addition, we find that SAM 2 is *more robust in crowded scenes* as measured by the decay metric defined in the VIPOSeg benchmark (lower is better; reflecting how fast the model's performance declines with more objects). On the VIPOSeg val split, SAM 2 has $\lambda = 0.68$ decay (without fine-tuning on VIPOSeg) and $\lambda = 0.67$ decay (when fine-tuned on the VIPOSeg training split), both outperforming the $\lambda = 0.70$ decay of the PAOT model.

| Method | training data | $\mathcal{G}$ | $\lambda$ (lower is better) |
|---|---|---|---|
| PAOT (Xu et al., 2023) | VIPOSeg + YouTube-VOS + DAVIS | 77.9 | 0.73 |
| PAOT (Xu et al., 2023) | VIPOSeg | 78.2 | 0.70 |
| **SAM 2** (zero-shot) | our mix | 78.4 | 0.68 |
| **SAM 2** (fine-tuned on VIPOSeg) | our mix + VIPOSeg | **79.7** | **0.67** |

Table 15: Performance of SAM 2 on the VIPOSeg benchmark. SAM 2 outperforms PAOT (the best model in Xu et al. (2023) trained on this dataset) under both zero-shot and fine-tuned settings.

## F.4 ZERO-SHOT IMAGE TASKS

### F.4.1 DATASET DETAILS

For the interactive segmentation task, we evaluated SAM 2 on a comprehensive suite of 37 datasets. This suite includes the 23 datasets previously used by SAM for zero-shot evaluation. For completeness, we list the 23 datasets: LVIS (Gupta et al., 2019), ADE20K (Zhou et al., 2019), Hypersim (Roberts et al., 2021), Cityscapes (Cordts et al., 2016), BBBC038v1 (Caicedo et al., 2019), DOORS (Pugliatti & Topputo, 2022), DRAM (Cohen et al., 2022), EgoHOS (Zhang et al., 2022), GTEA (Fathi et al., 2011; Li et al., 2015), iShape (Yang et al., 2021a), NDD20 (Trotter et al., 2020), NDISPark (Ciampi et al., 2021; 2022), OVIS (Qi et al., 2022), PPDLS (Minervini et al., 2016), Plittersdorf (Haucke et al., 2022), STREETS (Snyder & Do, 2019), TimberSeg (Fortin et al., 2022), TrashCan (Hong et al., 2020), VISOR (Darkhalil et al., 2022; Damen et al., 2022), WoodScape (Yogamani et al., 2019), PIDRay (Wang et al., 2021a), ZeroWaste-f (Bashkirova et al., 2022), and IBD (Chen et al., 2022). For more detailed information about these datasets, we refer the reader to Kirillov et al. (2023). In addition to these 23 datasets, we evaluated on frames sampled from 14 video datasets to assess SAM 2's performance on images from the video domain. The video datasets used are listed as follows: Lindenthal Camera Traps (LCT) (Haucke & Steinhage, 2021), VOST (Tokmakov et al., 2022), LV-VIS (Wang et al., 2023), FBMS (Brox et al., 2010), Virtual KITTI 2 (Cabon et al., 2020), Corsican Fire Database (CFD) (Toulouse et al., 2017), VIPSeg (Miao et al., 2022), Drosophila Heart OCM (DH OCM) (Fishman et al., 2023), EndoVis 2018 (Allan et al., 2020), ESD (Huang et al., 2023), UVO (Wang et al., 2021b), Ego-Exo4d (Grauman et al., 2023), LVOSv2 (Hong et al., 2024), and HT1080WT (Gómez-de Mariscal et al., 2021). Table 18 has a more detailed description of these datasets. (Some of these datasets are obtained from the same data source as the zero-shot video datasets in §F.1.1.)

### F.4.2 DETAILED ZERO-SHOT EXPERIMENTS

In this section, we include a more detailed version of the experiments in §6.3. We compare SAM 2 to SAM and HQ-SAM with different model sizes in Table 16. The main metrics we use for evaluation are the 1- and 5-click mIoU and we categorize the results by the dataset domain.

Table 16 first shows a comparison of the models trained only on images (for the SA task) with different image encoder sizes on both the SA-23 benchmark as well as the 14 newly introduced video datasets. SAM 2 (Hiera-B+) trained only on SA-1B outperforms SAM (ViT-H) on 1-click accuracy, and both SAM (ViT-H) and HQ-SAM (ViT-H) on 5-click accuracy while being 6x faster. SAM 2 (Hiera-L) further improves the 1-click accuracy by 1 point on average, but trading off speed. Despite being slower than Hiera-B+, it is still 3.4x faster than SAM (ViT-H) and 1.5x faster than SAM (ViT-B).

| Model | Data | 1 (5) click mIoU | | | | FPS |
| | | SA-23 All | SA-23 Image | SA-23 Video | 14 new Video | |
|---|---|---|---|---|---|---|
| SAM (ViT-B) | SA-1B | 55.9 (80.9) | 57.4 (81.3) | 54.0 (80.4) | 54.5 (82.6) | 76.7 |
| SAM (ViT-H) | SA-1B | 58.1 (81.3) | 60.8 (82.1) | 54.5 (80.3) | 59.1 (83.4) | 21.7 |
| HQ-SAM (ViT-B) | HQSEG-44k | 53.9 (72.1) | 56.3 (73.9) | 50.7 (69.9) | 54.5 (75.0) | 73.5 |
| HQ-SAM (ViT-H) | HQSEG-44k | 59.1 (79.8) | 61.8 (80.5) | 55.7 (78.9) | 58.9 (81.6) | 21.4 |
| **SAM 2** (Hiera-B+) | SA-1B | 58.9 (81.7) | 60.8 (82.1) | 56.4 (81.2) | 56.6 (83.7) | **130.1** |
| **SAM 2** (Hiera-L) | SA-1B | 60.0 (81.8) | 62.0 (82.2) | 57.4 (81.2) | 58.5 (83.8) | 61.4 |
| **SAM 2** (Hiera-B+) | our mix | 61.9 (**83.5**) | 63.3 (**83.8**) | 60.1 (**83.2**) | 69.6 (**85.8**) | **130.1** |
| **SAM 2** (Hiera-L) | our mix | **63.6** (83.5) | **64.7** (83.7) | **62.2** (83.2) | **71.1** (85.7) | 61.4 |

Table 16: Zero-shot performance on the Segment Anything (SA) task across a suite of 37 datasets. The table shows the average 1- and 5- click mIoU of SAM 2 compared to two baselines, categorized by dataset domain. We report the average metrics on the 23 datasets used by SAM for zero-shot evaluation, as well as the average across 14 newly introduced zero-shot video benchmarks.

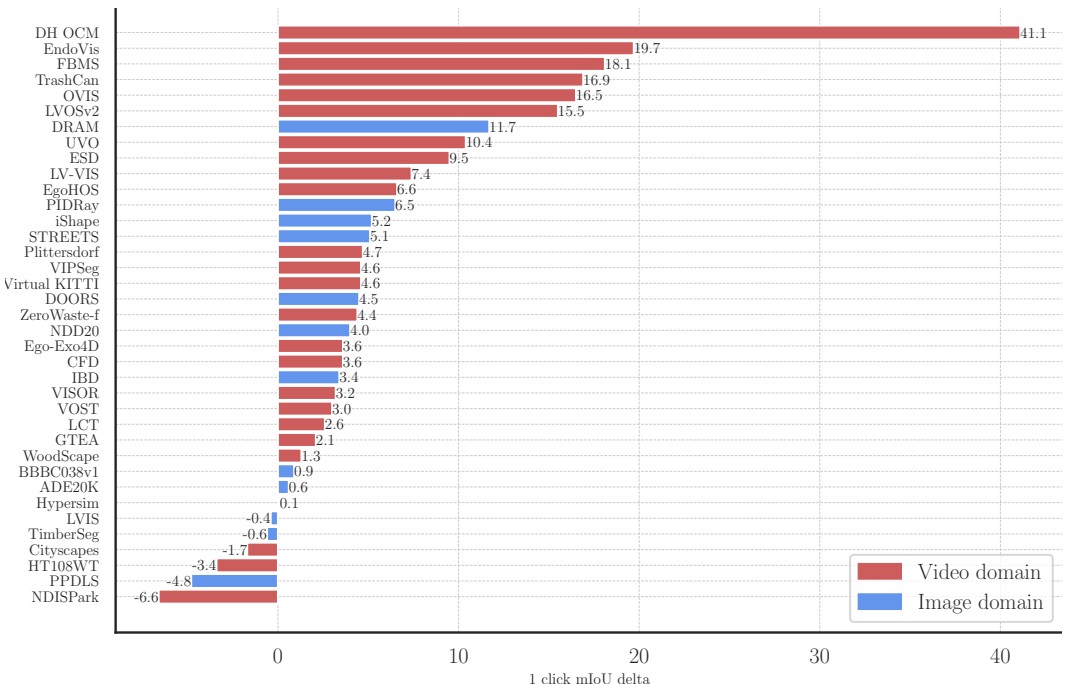

Figure 16: Zero-shot performance of SAM 2 vs SAM on a suite of 37 datasets. The figure shows the center 1 click mIoU delta between SAM 2 and SAM. Datasets derived from video distribution are highlighted in red, while those from image distribution are highlighted in blue.

The last two rows in Table 16 illustrate the benefits of training with our mix of image and video data, which boosts the average accuracy to 61.4% across the 23 datasets with the Hirea-B+ image encoder. Additionally, we observe substantial improvements on the video benchmarks of SA-23 as well as the 14 newly introduced video datasets. We note that we do not scale beyond Hiera-L, but expect better performance for a larger model.

A breakdown of the accuracy across datasets is presented in Fig. 16, where the per-dataset delta in 1-click mIoU relative to SAM is color-coded to indicate the data type (image or video). Notably, SAM 2 (Hiera-B+) surpasses SAM on 29 datasets[3] by up to 53.9 mIoU, despite using a smaller Hiera-B+ image encoder.

---

[3]We note that OVIS is not strictly zero-shot for SAM 2 since its videos are used in MOSE (part of our training data mix).

## G    APPLICATION OF SA-V DATA TO OTHER MODELS

The *segment anything in images and videos* capability of SAM 2 can be attributed to both its model design as well as its large-scale, high-quality training data, including SA-1B and SA-V.

Note that the SA-V dataset was collected in a model-in-the-loop fashion, in order to improve SAM 2 and *correct its failure cases*, while at the same time enabling flexible annotation for targeted data collection, including full objects and parts. This iterative process led to improvements in both the model and the data.

In this section, we aim to evaluate if the SA-V data can also benefit existing models, beyond SAM 2. To evaluate this, we use SA-V to train a recent semi-supervised VOS model, Cutie (Cheng et al., 2023a), which SA-V was *not* collected for, to assess SA-V's generalization capabilities.

To assess the generality of SA-V, we compare the performance of Cutie models trained with and without it, both for the existing VOS task with mask input and for interactive video segmentation with click or bounding box input (identical to the setup in §6.2). We report results in Table 17.

| Method | mask | | | | 5-click | box | mask |
| | SA-V test | MOSE val | DAVIS 2017 test | LVOS val | 17 zero-shot datasets (§6.2) | | |
| --- | --- | --- | --- | --- | --- | --- | --- |
| Cutie | 62.2 | 68.3 | 85.3 | 63.5 | 71.7 | 68.6 | 73.5 |
| Cutie + SA-V | 68.7 | 67.6 | 84.5 | 65.0 | 73.4 | 70.4 | 75.5 |
| SAM 2 | **78.4** | **77.9** | **87.7** | **78.0** | **77.6** | **74.4** | **79.3** |

Table 17: Our SA-V data has a broader utility beyond SAM 2 and helps zero-shot generalization in other models. Including SA-V in the training mix improves the performance of Cutie on SA-V test and provides zero-shot gains on LVOS and 17 zero-shot datasets. The final performance of Cutie + SA-V still falls short of the full SAM 2 shown in the last row.

As a baseline, the first row of Table 17 shows the $\mathcal{J}\&\mathcal{F}$ accuracy of the official Cutie model trained on DAVIS, YouTubeVOS, and MOSE (referred to as "w/ MOSE" in Cheng et al. (2023a)) as evaluated by us on SA-V test and our zero-shot benchmark of 17 datasets (using 5 clicks or bounding box as input to SAM+Cutie or ground-truth mask input to Cutie; see §F.1.3 for details). Performance on MOSE, DAVIS, and LVOS benchmarks is from officially reported numbers.

In the second row of Table 17, we train another Cutie model on our data mixture consisting of SA-V, DAVIS, YouTube-VOS, and MOSE. We use the same data mixture ratio as for SAM 2 and follow the official Cutie training recipe. Including SA-V in the training mix brings a large performance improvement on the SA-V benchmark. Although integrating SA-V into the training mix does not improve the performance on MOSE and DAVIS *in-domain* benchmarks, it boosts the *zero-shot* performance as shown by gains of +1.5 on LVOS and +2.0 on our 17 zero-shot benchmark suite under mask input, and similar boost for the out-of-domain click and box prompts.

Finally, the last row shows the accuracy of SAM 2, which significantly outperforms the Cutie models under both training settings, suggesting that the SAM 2 architecture design is another crucial aspect of the final performance.

## H    DETAILS ON COMPARISON TO STATE-OF-THE-ART IN SEMI-SUPERVISED VOS

We provide additional details on the comparison to the previous state-of-the-art in semi-supervised VOS (§7). We include results from SAM 2 trained only on SA-1B, SA-V and Internal data, for different encoder sizes.

**Qualitative comparison:** In Fig. 17, we show a comparison between our baseline (Cutie-base+, top row) and our model (SAM 2, bottom row) when prompted with a mask in the first frame. While the mask prompt in the first frame only covers the shirt of the person, the masklet predicted by the baseline wrongfully propagates to the whole person. Our model, however, is able to restrict the masklet to the target object.

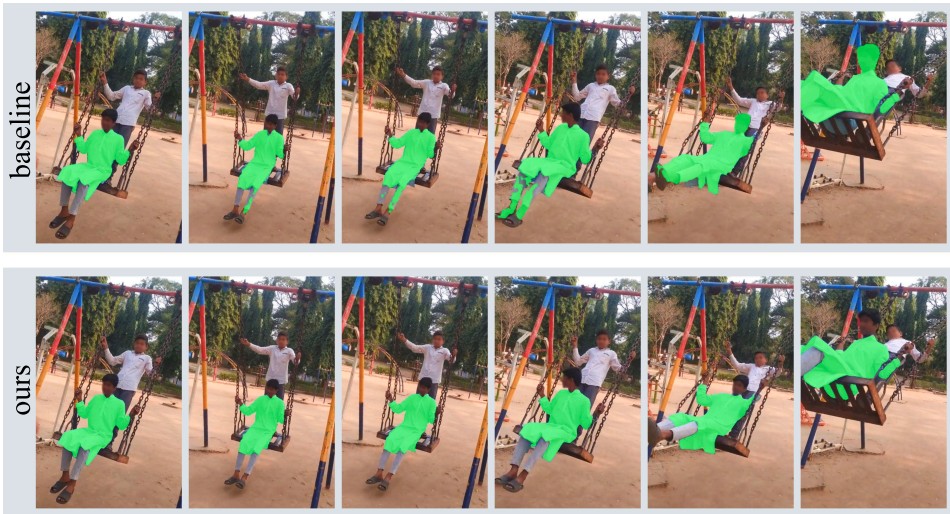

Figure 17: Comparison between our baseline (Cutie-base+, top row) and our model (SAM 2, bottom row) when prompted with a mask in the first frame.

| dataset | abbreviation & link | video type | description | annotation type | source split | # videos sampled | # masklets sampled | # frames sampled | # masks sampled |
|---|---|---|---|---|---|---|---|---|---|
| LV-VIS (Wang et al., 2023) | LV-VIS | Open Vocabulary | Large scale open vocabulary video instance segmentation | Dense | Validation | 690 | 2536 | 15,604 | 54,077 |
| A Drosophila heart optical coherence microscopy database for automatic video segmentation (Fishman et al., 2023) | DH OCM | Microscopy; heart | Segmentation of a fruit fly heart in optical coherence microscopy videos | Sparse | All | 213 | 213 | 608,000 | 607,158 |
| Video Object Segmentation under Transformations (Tokmakov et al., 2022) | VOST | Deformation | Video object segmentation with emphasis on shape transformations | Dense | Validation | 635 | 882 | 67,004 | 89,722 |
| Cityscapes-VPS (Cordts et al., 2016; Kim et al., 2020b) | Cityscapes-VPS | Driving | Panoptic segmentation for Cityscapes driving dataset | Sparse | Validation Instance | 209 | 1372 | 4,259 | 4,579 |
| Corsican Fire Database (Toulouse et al., 2017) | CFD | Wildfire | Segmentation of wildfires | Sparse | All | 5 | 5 | 541 | 541 |
| Partial and Unusual Masks for Video Object Segmentation (Bekuzarov et al., 2023) | PUMaVOS | Parts | Video object segmentation with a focus on parts and practical use cases | Dense | All | 24 | 26 | 21,187 | 21,485 |
| EPIC-KITCHENS VISOR (Darkhalil et al., 2022) | VISOR | Egocentric | Video object segmentation benchmark containing egocentric videos of cooking with an emphasis on segmenting active objects. | Sparse | Validation | 921 | 921 | 736,030 | 4,426 |
| HT1080WT cells embedded in 3D collagen type I matrices (Gómez-de Mariscal et al., 2021) | HT1080WT | Microscopy; cells | Timelapse videos of HT1080WT cell movement | Sparse | All | 60 | 150 | 1,010 | 2,694 |
| Freiburg-Berkeley Motion Segmentation Dataset (Brox et al., 2010) | FBMS | Moving Object | Precise segmentation of moving objects | Sparse | All | 45 | 70 | 9734 | 755 |
| Virtual KITTI 2 (Cabon et al., 2020) | Virtual KITTI 2 | Synthetic; Driving | Synthetic driving videos generated by a game engine that recreate real world KITTI videos. | Dense | All from video angle Camera_0 | 996 | 1,638 | 109,368 | 162,708 |
| EndoVis 2018 (Allan et al., 2020) | EndoVis 2018 | Endoscopic video; surgery | Segmentation of medical tools in endoscopic videos | Dense | All | 15 | 29 | 2,325 | 4,314 |
| Lindenthal Camera Traps (Haucke & Steinhage, 2021) | LCT | Stereo | Wildlife videos captured using stereo cameras. | Sparse | All | 12 | 12 | 4,012 | 412 |
| LVOSv2 (Hong et al., 2024) | LVOSv2 | Long videos | Long-term video object segmentation benchmark, on average 1.14 minutes | Dense | Validation | 136 | 225 | 64,523 | 91,510 |
| UVO (Wang et al., 2021b) | UVO | Open World | Open World instance segmentation of all objects in a video | Dense | Validation | 54 | 311 | 4,860 | 26,747 |
| EgoExo4d (Grauman et al., 2023) | EgoExo4d | Egocentric | Egocentric videos of participants completing skilled activities. | Sparse | Validation videos on egocentric cameras | 1185 | 1185 | 327,080 | 9,035 |
| VIPSeg (Miao et al., 2022) | VIPSeg | Panoptic | Large scale and real world scenarios for video panoptic segmentation | Dense | Validation | 152 | 1,457 | 3,416 | 30,408 |
| Event-based Segmentation Dataset (Huang et al., 2023) | ESD | Clutter | Tabletop object segmentation in an indoor cluttered environment | Dense | All | 135 | 814 | 13,325 | 78,243 |

Table 18: Video segmentation datasets used for zero-shot evaluation.

**Quantitative comparison:** In Table 19, we compare the performance of our model to previous approaches on additional semi-supervised VOS metrics. SAM 2 outperforms prior work on all evaluated benchmarks, in all metrics. Note that unlike these previous approaches, SAM 2 is not specialized in the semi-supervised VOS task but is capable of more general promptable segmentation. SAM 2 is also not restricted to a specific set of object classes. The performance of our model on the SA-V benchmark (Table 19a) demonstrates its capability to segment anything in a video.

# I    OTHER RELATED WORK

We mention other related works that tackle tasks which are not our primary focus, but are highly relevant as these share similarities with our work.

**Video Instance Segmentation (VIS).** The VIS (Yang et al., 2019) task focuses on simultaneous detection, segmentation and tracking of instances in videos. Some notable works in VIS include the use of transformer based architectures (Wang et al., 2021c; Li et al., 2023b; Lee et al., 2024). Another line of work (Zhang et al., 2023c;d) focuses on decoupling the VIS task into three sub-tasks: segmentation, tracking, and refinement.

**Video Panoptic Segmentation (VPS).** The VPS (Kim et al., 2020a) task requires segmenting both things and stuff as well as associating instances across frames of the video. Recent works (Shin et al., 2024; Li et al., 2022b; 2024) aim to tackle this task, and they have shown promising results on benchmark datasets (Miao et al., 2022).

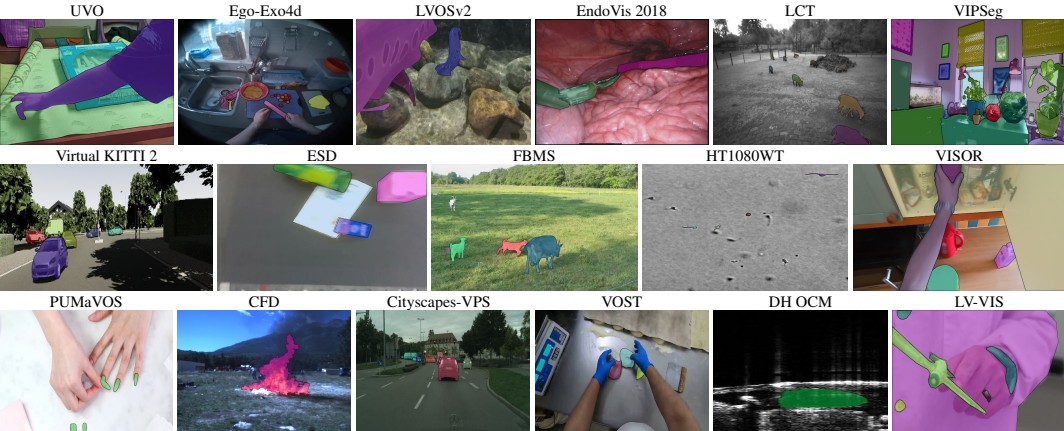

Figure 18: Examples from SAM 2 zero-shot video benchmark suite.

| Method | SA-V val $\mathcal{J}\&\mathcal{F}$ | $\mathcal{J}$ | $\mathcal{F}$ | SA-V test $\mathcal{J}\&\mathcal{F}$ | $\mathcal{J}$ | $\mathcal{F}$ |
|---|---|---|---|---|---|---|
| STCN (Cheng et al., 2021a) | 61.0 | 57.4 | 64.5 | 62.5 | 59.0 | 66.0 |
| SwinB-AOT-L (Yang et al., 2021b) | 51.1 | 46.4 | 55.7 | 50.3 | 46.0 | 54.6 |
| SwinB-DeAOT-L (Yang & Yang, 2022) | 61.4 | 56.6 | 66.2 | 61.8 | 57.2 | 66.3 |
| RDE (Li et al., 2022a) | 51.8 | 48.4 | 55.2 | 53.9 | 50.5 | 57.3 |
| XMem (Cheng & Schwing, 2022) | 60.1 | 56.3 | 63.9 | 62.3 | 58.9 | 65.8 |
| SimVOS-B (Wu et al., 2023b) | 44.2 | 40.0 | 48.3 | 44.1 | 40.5 | 47.7 |
| DEVA (Cheng et al., 2023b) | 55.4 | 51.5 | 59.2 | 56.2 | 52.4 | 60.1 |
| Cutie-base (Cheng et al., 2023a) | 60.7 | 57.7 | 63.7 | 62.7 | 59.7 | 65.7 |
| Cutie-base+ (Cheng et al., 2023a) | 61.3 | 58.3 | 64.4 | 62.8 | 59.8 | 65.8 |
| SAM 2 (Hiera-B+) | 76.8 | 73.4 | 80.1 | 77.0 | 73.5 | 80.5 |
| SAM 2 (Hiera-L) | 77.9 | 74.5 | 81.3 | 78.4 | 74.9 | 82.0 |
| SAM 2 (Hiera-T)‡ | 75.2 | 71.7 | 78.7 | 76.5 | 72.9 | 80.1 |
| SAM 2 (Hiera-S)‡ | 77.0 | 73.6 | 80.5 | 76.6 | 73.0 | 80.2 |
| SAM 2 (Hiera-B+)‡ | 77.5 | 74.1 | 80.9 | 78.2 | 74.8 | 81.7 |
| SAM 2 (Hiera-L)‡ | **78.6** | **75.3** | **82.0** | **79.5** | **76.0** | **83.0** |

(a) Comparisons between SAM 2 and previous work on our SA-V benchmark for the semi-supervised VOS task. We evaluated prior works on SA-V using their open-sourced code and checkpoints.

| Method | LVOS val $\mathcal{J}\&\mathcal{F}$ | $\mathcal{J}$ | $\mathcal{F}$ |
|---|---|---|---|
| DEVA (Cheng et al., 2023b) | 55.9 | 51.1 | 60.7 |
| DDMemory (Hong et al., 2023) | 60.7 | 55.0 | 66.3 |
| Cutie-base (Cheng et al., 2023a) | 66.0 | 61.3 | 70.6 |
| SAM 2 (Hiera-B+) | 78.0 | 73.2 | 82.7 |
| SAM 2 (Hiera-L) | 78.0 | 73.2 | 82.7 |
| SAM 2 (Hiera-T)‡ | 77.5 | 73.0 | 82.1 |
| SAM 2 (Hiera-S)‡ | 77.3 | 72.3 | 82.2 |
| SAM 2 (Hiera-B+)‡ | 77.7 | 73.1 | 82.4 |
| SAM 2 (Hiera-L)‡ | **80.1** | **75.4** | **84.9** |

| Method | LVOSv2 val $\mathcal{J}\&\mathcal{F}$ | $\mathcal{J}_s$ | $\mathcal{F}_s$ | $\mathcal{J}_u$ | $\mathcal{F}_u$ |
|---|---|---|---|---|---|
| STCN (Cheng et al., 2021a) | 60.6 | 57.2 | 64.0 | 57.5 | 63.8 |
| RDE (Li et al., 2022a) | 62.2 | 56.7 | 64.1 | 60.8 | 67.2 |
| SwinB-DeAOT-L (Yang & Yang, 2022) | 63.9 | 61.5 | 69.0 | 58.4 | 66.6 |
| XMem (Cheng & Schwing, 2022) | 64.5 | 62.6 | 69.1 | 60.6 | 65.6 |
| SAM 2 (Hiera-B+) | 78.7 | 80.6 | 87.2 | 69.0 | 77.8 |
| SAM 2 (Hiera-L) | 79.6 | 81.0 | 87.4 | 70.4 | 79.8 |
| SAM 2 (Hiera-T)‡ | 77.3 | 79.9 | 86.9 | 67.1 | 75.4 |
| SAM 2 (Hiera-S)‡ | 78.3 | 79.2 | 85.9 | 69.0 | 79.0 |
| SAM 2 (Hiera-B+)‡ | 78.2 | 80.5 | 87.2 | 68.1 | 76.9 |
| SAM 2 (Hiera-L)‡ | **80.6** | **81.7** | **88.2** | **71.4** | **81.0** |

(b) Comparisons between SAM 2 and previous work on the LVOS (Hong et al., 2023) benchmark.

(c) Comparisons between SAM 2 and previous work on the LVOSv2 (Hong et al., 2024) benchmark. We report the performance of prior works as evaluated by the LVOSv2 authors.

| Method | MOSE val $\mathcal{J}\&\mathcal{F}$ | $\mathcal{J}$ | $\mathcal{F}$ | DAVIS17 val $\mathcal{J}\&\mathcal{F}$ | $\mathcal{J}$ | $\mathcal{F}$ | DAVIS17 test $\mathcal{J}\&\mathcal{F}$ | $\mathcal{J}$ | $\mathcal{F}$ | YTVOS19 val $\mathcal{G}$ | $\mathcal{J}_s$ | $\mathcal{F}_s$ | $\mathcal{J}_u$ | $\mathcal{F}_u$ |
|---|---|---|---|---|---|---|---|---|---|---|---|---|---|---|
| STCN (Cheng et al., 2021a) | 52.5 | 48.5 | 56.6 | 85.4 | 82.2 | 88.6 | 76.1 | 72.7 | 79.6 | 82.7 | 81.1 | 85.4 | 78.2 | 85.9 |
| SwinB-AOT-L (Yang et al., 2021b) | 59.4 | 55.5 | 63.2 | 85.4 | 82.4 | 88.4 | 81.2 | 77.3 | 85.1 | 84.5 | 84.0 | 88.8 | 78.4 | 86.7 |
| SwinB-DeAOT-L (Yang & Yang, 2022) | 59.9 | 55.7 | 64.0 | 86.2 | 83.1 | 89.2 | 82.8 | 78.9 | 86.7 | 86.1 | 85.3 | 90.2 | 80.4 | 88.6 |
| RDE (Li et al., 2022a) | 46.8 | 42.4 | 51.3 | 84.2 | 80.8 | 87.5 | 77.4 | 73.6 | 81.2 | 81.9 | 81.1 | 85.5 | 76.2 | 84.8 |
| XMem (Cheng & Schwing, 2022) | 59.6 | 55.4 | 63.7 | 86.0 | 82.8 | 89.2 | 79.6 | 76.1 | 83.0 | 85.6 | 84.1 | 88.5 | 81.0 | 88.9 |
| SimVOS-B (Wu et al., 2023b) | - | - | - | 88.0 | 85.0 | 91.0 | 80.4 | 76.1 | 84.6 | 84.2 | 83.1 | - | 79.1 | - |
| JointFormer (Zhang et al., 2023b) | - | - | - | 90.1 | 87.0 | 93.2 | 88.1 | 84.7 | 91.6 | 87.4 | 86.5 | 90.9 | 82.0 | 90.3 |
| ISVOS (Wang et al., 2022) | - | - | - | 88.2 | 84.5 | 91.9 | 84.0 | 80.1 | 87.8 | 86.3 | 85.2 | 89.7 | 81.0 | 89.1 |
| DEVA (Cheng et al., 2023b) | 66.0 | 61.8 | 70.3 | 87.0 | 83.8 | 90.2 | 82.6 | 78.9 | 86.4 | 85.4 | 84.9 | 89.4 | 79.6 | 87.8 |
| Cutie-base (Cheng et al., 2023a) | 69.9 | 65.8 | 74.1 | 87.9 | 84.6 | 91.1 | 86.1 | 82.4 | 89.9 | 87.0 | 86.0 | 90.5 | 82.0 | 89.6 |
| Cutie-base+ (Cheng et al., 2023a) | 71.7 | 67.6 | 75.8 | 88.1 | 85.5 | 90.8 | 88.1 | 84.7 | 91.4 | 87.5 | 86.3 | 90.6 | 82.7 | 90.5 |
| SAM 2 (Hiera-B+) | 76.6 | 72.6 | 80.6 | 90.2 | 87.0 | 93.4 | 87.9 | 84.7 | 91.1 | 88.6 | 87.1 | 91.6 | 83.9 | 91.9 |
| SAM 2 (Hiera-L) | **77.9** | **73.9** | **81.9** | 90.7 | 87.5 | 94.0 | 87.7 | 84.6 | 90.9 | 89.3 | 87.5 | 92.0 | 84.8 | 92.8 |
| SAM 2 (Hiera-T)‡ | 71.8 | 67.4 | 76.1 | 89.4 | 85.8 | 92.9 | 86.9 | 83.4 | 90.3 | 87.4 | 85.7 | 90.1 | 82.7 | 91.1 |
| SAM 2 (Hiera-S)‡ | 73.5 | 69.2 | 77.7 | 89.6 | 86.3 | 92.9 | 87.6 | 84.1 | 91.1 | 88.0 | 85.9 | 90.2 | 83.9 | 91.9 |
| SAM 2 (Hiera-B+)‡ | 73.8 | 69.7 | 77.9 | 90.0 | 86.8 | 93.1 | 86.6 | 83.3 | 89.8 | 88.2 | 86.1 | 90.6 | 84.0 | 92.1 |
| SAM 2 (Hiera-L)‡ | 74.6 | 70.6 | 78.6 | 90.2 | 86.9 | 93.4 | **88.9** | **85.3** | **92.5** | 88.8 | 86.5 | 91.0 | 84.7 | 92.8 |

(d) Comparisons between SAM 2 and previous work on the semi-supervised VOS task.

Table 19: Detailed comparisons between SAM 2 and previous work on various benchmarks (‡: a version of the model trained on SA-1B, SA-V, and our internal dataset as described in §5.2).

# J MODEL, DATA AND ANNOTATION CARDS

## J.1 MODEL CARD

**Model Overview**

| | |
|---|---|
| Name | SAM 2 (Segment Anything Model 2) |
| Version | 1.0 |
| Date | 2024 |
| Organization | Meta FAIR |
| Mode type | Promptable segmentation model |
| Architecture | See Section 4 |
| Repository | `https://github.com/facebookresearch/sam2` |
| License | Apache 2.0 |

**Intended Use**

| | |
|---|---|
| Primary intended users | SAM 2 was designed as a unified model for promptable video and image segmentation tasks. The model was primarily developed for research use cases. SAM 2 is released under an Apache 2.0 license. |
| Out-of-scope use cases | See Ethical considerations and license for restrictions. |
| Caveats and recommendations | See Appendix C for limitations. |

**Relevant Factors**

| | |
|---|---|
| Groups | SAM 2 is class agnostic and was designed for promptable image and video segmentation. It can segment and track any object. |
| Instrumentation and environment | SAM 2 was evaluated across a variety of types of video and image data. The video benchmark suite included domains such as *driving data, microscopy, egocentric video, robotic surgery*. See Table 18 for descriptions of the benchmarks and Figure 18 for example frames. SAM 2 was evaluated on the same suite of image benchmarks as Kirillov et al. (2023), which covers domains including *underwater images, paintings, fish-eye images*. |

**Metrics**

| | |
|---|---|
| | We evaluate the performance of SAM 2 using the following metrics: |
| | $\mathcal{J}\&\mathcal{F}$: We evaluate performance using $\mathcal{J}\&\mathcal{F}$ (Pont-Tuset et al., 2017) for the promptable video segmentation and semi-supervised VOS tasks. |
| | $\mathcal{G}$: We use $\mathcal{G}$ for evaluation on YTVOS 2019 for the semi-supervised VOS task. |
| | *mIoU*: We evaluate performance using mIoU for the promptable image segmentation task. |

**Evaluation Data**

| | |
|---|---|
| Data sources | See Appendix F |

**Training Data**

| | |
|---|---|
| Data source | SAM 2 was trained on the SA-V dataset alongside internally available licensed video data. See Section 5 of the main text for more details and Appendix J.2 for the SA-V dataset data card. |

**Ethical Considerations**

| | |
|---|---|
| Data | See Section 5 for more details about the SAM 2 training data. In Section E.1 we show a geographic distribution of the videos and demographic distribution of the crowdworkers who collected the videos in the SA-V dataset. |
| Cost and impact of compute | The released SAM 2 was trained on 256 A100 GPUs for 108 hours. This corresponds to 12165.12 kWH and an estimated emissions of 3.89 metric tons of CO2e (Patterson et al., 2021; Lacoste et al., 2019). The emissions from training the released SAM 2 are equivalent to ∼10k miles driven by an average gasoline-powered passenger vehicle (Agency, 2022). |
| Risks and harms | In Section E.1.1 of the main text we analyze SAM 2 performance on people across demographic groups. When using SAM 2 in new settings, we suggest that researchers perform their own fairness evaluation for SAM 2 specific to their use case. |
| Use cases | We implore users to use their best judgement. |

Table 20: Model card for SAM 2 following the structure in Mitchell et al. (2019)

## J.2 DATASET CARD FOR SA-V DATASET

**Motivation**

1. *For what purpose was the dataset created? Was there a specific task in mind? Was there a specific gap that needed to be filled? Please provide a description.* The dataset was designed for the PVS task. The contributions of our dataset to the vision community are: (1) The dataset, composed of 50.9K videos and 642.6K masklets, is the largest video segmentation dataset publicly available today (see 5.2 for comparisons to current VOS datasets) (2) The dataset is available under a Creative Commons Attribution 4.0 International Public License at https://ai.meta.com/datasets/segment-anything-video/, (3) The data is a more geographically diverse, publicly available, video segmentation dataset than its predecessors.

2. *Who created the dataset (e.g., which team, research group) and on behalf of which entity (e.g., company, institution, organization)?* The dataset was created by Meta FAIR. The underlying videos were collected via a contracted third party company.

3. *Who funded the creation of the dataset?* The dataset was funded by Meta FAIR.

4. *Any other comments?* No.

**Composition**

1. *What do the instances that comprise the dataset represent (e.g., documents, photos, people, countries)? Are there multiple types of instances (e.g., movies, users, and ratings; people and interactions between them; nodes and edges)? Please provide a description.* All of the instances in the dataset are videos. Subject matter diversity was encouraged and no specific themes were applied during video collection. Common themes of the video include: locations, objects, scenes. All the videos are distinct, however there are some sets of videos that were taken of the same subject matter.

2. *How many instances are there in total (of each type, if appropriate)?* There are 50.9K videos.

3. *Does the dataset contain all possible instances or is it a sample (not necessarily random) of instances from a larger set? If the dataset is a sample, then what is the larger set? Is the sample representative of the larger set (e.g., geographic coverage)? If so, please describe how this representativeness was validated/verified. If it is not representative of the larger set, please describe why not (e.g., to cover a more diverse range of instances, because instances were withheld or unavailable).* While the dataset contains all possible instances, reviewers were advised to refuse to annotate content containing explicit imagery.

4. *What data does each instance consist of? "Raw" data (e.g., unprocessed text or images) or features? In either case, please provide a description.* Each instance is a video.

5. *Is there a label or target associated with each instance? If so, please provide a description.* Each video is annotated with masklets that track objects throughout the video. There are no categories or text associated with the masklets. The data was annotated at 6 FPS. There are an average of 3.8 manual masklets, and 8.9 auto masklets per video, and there are 642.6K masklets in total.

6. *Is any information missing from individual instances? If so, please provide a description, explaining why this information is missing (e.g., because it was unavailable). This does not include intentionally removed information, but might include, e.g., redacted text.* No.

7. *Are relationships between individual instances made explicit (e.g., users' movie ratings, social network links)? If so, please describe how these relationships are made explicit.* No.

8. *Are there any errors, sources of noise, or redundancies in the dataset? If so, please provide a description.* For manual masklets, human errors may exist; for example, annotators may miss a frame to check or fix when needed. For auto masklets, as SAM 2 is used to generates them, model errors such as inconsistencies in the masklets may exist.

9. *Is the dataset self-contained, or does it link to or otherwise rely on external resources (e.g., websites, tweets, other datasets)? If it links to or relies on external resources, a) are there guarantees that they will exist, and remain constant, over time; b) are there official archival versions of the complete dataset (e.g., including the external resources as they existed at the time the dataset was created); c) are there any restrictions (e.g., licenses, fees) associated with any of the external resources that might apply to a dataset consumer? Please provide descriptions of all external resources and any restrictions associated with them, as well as links or other access points, as appropriate.* The dataset is self contained.

10. *Does the dataset contain data that might be considered confidential (e.g., data that is protected by legal privilege or by doctor-patient confidentiality, data that includes the content of individuals' non-public communications)? If so, please provide a description.* No.

11. *Does the dataset contain data that, if viewed directly, might be offensive, insulting, threatening, or might otherwise cause anxiety? If so, please describe why.* We have three safety measures to prevent objectionable content: (1) The video collecting crowdworkers were provided instructions to not record videos that might contain objectionable content (e.g., graphic, nudity, or inappropriate content). (2) The expert annotators who annotated the videos were provided instructions to flag and reject videos if objectionable content was present. (3) reports about video(s) in the dataset can be submitted to segment-anything@meta.com.

12. *Does the dataset identify any subpopulations (e.g., by age, gender)? If so, please describe how these subpopulations are identified and provide a description of their respective distributions within the dataset.* The dataset does not identify any subpopulations of the people in the videos. The demographics of the crowdworkers who collected the videos in the dataset are presented in 5.2.

13. *Is it possible to identify individuals (i.e, one or more natural persons), either directly or indirectly (i.e., in combination with other data) from the dataset? If so, please describe how.* Videos were subjected to a face blurring model. Reports about videos in the dataset can be submitted to segment-anything@meta.com.

14. *Does the dataset contain data that might be considered sensitive in any way (e.g., data that reveals race or ethnic origins, sexual orientations, religious beliefs, political opinions or union memberships, or locations; financial or health data; biometric or genetic data; forms of government identification, such as social security numbers; criminal history)? If so, please provide a description.* The dataset is not focused on data that may be considered sensitive. Reports about videos in the dataset can be submitted to segment-anything@meta.com.

15. *Any other comments?* No.

**Collection Process**

1. *How was the data associated with each instance acquired? Was the data directly observable (e.g., raw text, movie ratings), reported by subjects (e.g., survey responses), or indirectly inferred/derived from other data (e.g., part-of-speech tags, model-based*

*guesses for age or language)? If the data was reported by subjects or indirectly inferred/derived from other data, was the data validated/verified? If so, please describe how.* The released masklets associated with each video were collected using two methods. (1) SAM 2 assisted manual annotation (2) automatically generated by SAM 2 and verified by annotators.

2. *What mechanisms or procedures were used to collect the data (e.g., hardware apparatuses or sensors, manual human curation, software programs, software APIs)? How were these mechanisms or procedures validated?* The videos in the dataset were collected via a contracted third-party vendor. They are videos taken by crowdworkers with unknown equipment.

3. *If the dataset is a sample from a larger set, what was the sampling strategy (e.g., deterministic, probabilistic with specific sampling probabilities)?* N/A.

4. *Who was involved in the data collection process (e.g., students, crowdworkers, contractors) and how were they compensated (e.g., how much were crowdworkers paid)?* (1) The videos in the dataset were collected via a contracted third-party vendor. They are videos taken by crowdworkers who were compensated with an hourly wage set by the vendor. (2) The manually collected masklets in the dataset were collected by annotators via another third-party vendor. Annotators were compensated with an hourly wage set by the vendor.

5. *Over what timeframe was the data collected? Does this timeframe match the creation timeframe of the data associated with the instances (e.g., recent crawl of old news articles)? If not, please describe the timeframe in which the data associated with the instances was created.* The videos were filmed between November 2023 and March 2024. The masklet annotations were collected between April 2024 and July 2024.

6. *Were any ethical review processes conducted (e.g., by an institutional review board)? If so, please provide a description of these review processes, including the outcomes, as well as a link or other access point to any supporting documentation. If the dataset does not relate to people, you may skip the remaining questions in this section.* The project underwent an internal review process.

7. *Did you collect the data from the individuals in question directly, or obtain it via third parties or other sources (e.g. websites)?* We contracted with third-party vendors to collect the videos and to generate or review annotations.

8. *Were the individuals in question notified about the data collection? If so, please describe (or show with screenshots or other information) how notice was provided, and provide a link or other access point to, or otherwise reproduce, the exact language of the notification itself.* The videos were collected by crowdworkers via a contracted third-party vendor. The crowdworkers agreed to consent forms.

9. *Did the individuals in question consent to the collection and use of their data? If so, please describe (or show with screenshots or other information) how consent was requested and provided, and provide a link or other access point to, or otherwise reproduce, the exact language to which the individuals consented.* The videos were collected via a contracted third-party who provided appropriate representations regarding the collection of any notices and consents as required from individuals.

10. *If consent was obtained, were the consenting individuals provided with a mechanism to revoke their consent in the future or for certain uses? If so, please provide a description, as well as a link or other access point to the mechanism (if appropriate).* Pursuant to the contract, the contracted third-party collected consents and provided opportunity for consent revocation.

11. *Has an analysis of the potential impact of the dataset and its use on data subjects (e.g., a data protection impact analysis) been conducted? If so, please provide a description of this analysis, including the outcomes, as well as a link or other access point to any supporting documentation.* See detail in E.1.1.

12. *Any other comments?* No.

### Preprocessing / Cleaning / Labeling

1. *Was any preprocessing / cleaning / labeling of the data done (e.g., discretization or bucketing, tokenization, part-of-speech tagging, SIFT feature extraction, removal of instances, processing of missing values)? If so, please provide a description. If not, you may skip the remaining questions in this section.* The videos were re-sampled to 24 fps and converted to mp4 format.

2. *Was the "raw" data saved in addition to the preprocessed/cleaned/labeled data (e.g., to support unanticipated future uses)? If so, please provide a link or other access point to the "raw" data.* No.

### Uses

1. *Has the dataset been used for any tasks already? If so, please provide a description.* The dataset has been used to train and evaluate SAM 2.

2. *What (other) tasks could the dataset be used for?* The data could be used for VOS, iVOS, or PVS tasks. If frames are sampled from the videos, the dataset can be used for the image segmentation task.

3. *Is there anything about the composition of the dataset or the way it was collected and preprocessed/cleaned/labeled that might impact future uses? For example, is there anything that a dataset consumer might need to know to avoid uses that could result in unfair treatment of individuals or groups (e.g., stereotyping, quality of service issues) or other risks or harms (e.g., legal risks, financial harms)? If so, please provide a description. Is there anything a dataset consumer could do to mitigate these risks or harms?* We have an analysis of the geography and crowdworker demographic of our dataset in 5.2. While we believe our dataset to be more representative on these factors than most of the publicly existing datasets of its kind at this time, we acknowledge that we do not have parity across all geographic and demographic groups, and we encourage users of the dataset to be mindful of any potential biases models may learn using this dataset.

4. *Are there tasks for which the dataset should not be used? If so, please provide a description.* No. Full terms of use for the dataset can be found at https://ai.meta.com/datasets/segment-anything-video-downloads/.

5. *Any other comments?* No.

### Distribution

1. *Will the dataset be distributed to third parties outside of the entity (e.g., company, institution, organization) on behalf of which the dataset was created? If so, please provide a description.* The dataset will be available under the permissive Creative Commons Attribution 4.0 International Public License.

2. *How will the dataset will be distributed (e.g., tarball on website, API, GitHub)? Does the dataset have a digital object identifier (DOI)?* The dataset is available at https://ai.meta.com/datasets/segment-anything-video/.

3. *When will the dataset be distributed?* The dataset will be distributed in July 2024.

4. *Will the dataset be distributed under a copyright or other intellectual property (IP) license, and/or under applicable terms of use (ToU)? If so, please describe this license and/or ToU, and provide a link or other access point to, or otherwise reproduce, any relevant licensing terms or ToU, as well as any fees associated with these restrictions.* Yes, the dataset will be available under the Creative Commons Attribution 4.0 International Public License. The license agreement and terms of use for the dataset can be found at https://ai.meta.com/datasets/segment-anything-video-downloads/. Users must agree to the terms of use before downloading or using the dataset.

5. *Have any third parties imposed IP-based or other restrictions on the data associated with the instances? If so, please describe these restrictions, and provide a link or other access point to, or otherwise reproduce, any relevant licensing terms, as well as any fees associated with these restrictions.* Full terms of use and restrictions on use of the SA-V dataset can be found at https://ai.meta.com/datasets/segment-anything-video-downloads/.

6. *Do any export controls or other regulatory restrictions apply to the dataset or to individual instances? If so, please describe these restrictions, and provide a link or other access point to, or otherwise reproduce, any supporting documentation.* The license and restrictions on use of the SA-V dataset can be found at https://ai.meta.com/datasets/segment-anything-video-downloads/.

7. *Any other comments?* No.

**Maintenance**

1. *Who will be supporting/hosting/maintaining the dataset?* The dataset will be hosted at https://ai.meta.com/datasets/segment-anything-video/ and maintained by Meta FAIR.

2. *How can the owner/curator/manager of the dataset be contacted (e.g., email address)?* Please email segment-anything@meta.com.

3. *Is there an erratum? If so, please provide a link or other access point.* No.

4. *Will the dataset be updated (e.g., to correct labeling errors, add new instances, delete instances)? If so, please describe how often, by whom, and how updates will be communicated to dataset consumers (e.g., mailing list, GitHub)?* Updates may be made pursuant to inbound received at segment-anything@meta.com.

5. *If the dataset relates to people, are there applicable limits on the retention of the data associated with the instances (e.g., were the individuals in question told that their data would be retained for a fixed period of time and then deleted)? If so, please describe these limits and explain how they will be enforced.* There are no limits on data retention.

6. *Will older versions of the dataset continue to be supported/hosted/maintained? If so, please describe how. If not, please describe how its obsolescence will be communicated to dataset consumers.* No. If updates are made to the dataset, previous versions will not continue to be hosted.

7. *If others want to extend/augment/build on/contribute to the dataset, is there a mechanism for them to do so? If so, please provide a description. Will these contributions be validated/verified? If so, please describe how. If not, why not? Is there a process for communicating/distributing these contributions to dataset consumers? If so, please provide a description.* We encourage further annotations for SA-V, but these will not be validated/verified or supported/hosted/maintained by Meta.

8. *Any other comments?* No.

## J.3 DATA ANNOTATION CARD

**Task Formulation**

1. *At a high level, what are the subjective aspects of your task?* Selecting objects to mask and track in a video is inherently a subjective task, and annotators might differ in their decision to mask objects.

2. *What assumptions do you make about annotators?* We assume our annotators understand the PVS task and are well trained on video related tasks. Our annotators worked full time on our annotation task. This made it possible to train the annotators by sharing feedback on a regular basis.

3. *How did you choose the specific wording of your task instructions? What steps, if any, were taken to verify the clarity of task instructions and wording for annotators?* (1) The task instructions included visual examples (images and videos) to provide clarity. (2) Annotators were well trained before working on production queues. (3) The research team shared feedback daily and met with the annotators weekly for Q&A sessions.

4. *What, if any, risks did your task pose for annotators and were they informed of the risks prior to engagement with the task?* Annotators were informed to reject objectionable videos.

5. *What are the precise instructions that were provided to annotators?* See detail in 12 for annotation instructions.

**Selecting Annotations**

1. *Are there certain perspectives that should be privileged? If so, how did you seek these perspectives out?* We chose to work with annotators with previous video annotation experience.

2. *Are there certain perspectives that would be harmful to include? If so, how did you screen these perspectives out?* No.

3. *Were sociodemographic characteristics used to select annotators for your task? If so, please detail the process.* For masklet annotations, sociodemographic characteristics were not used to select the annotators. For video collection, we emphasized the importance of diversity among the crowdworkers to our third-party vendor. While it was not a strict requirement, we encouraged the inclusion of a diverse group of crowdworkers to enrich the data collection process with a wide range of perspectives. This approach aimed to naturally incorporate diversity without imposing strict selection based on sociodemographic factors.

4. *If you have any aggregated socio-demographic statistics about your annotator pool, please describe. Do you have reason to believe that sociodemographic characteristics of annotators may have impacted how they annotated the data? Why or why not?* Aggregated socio-demographic statistics about the crowdworkers who collected the videos are presented in 5.2.

5. *Consider the intended context of use of the dataset and the individuals and communities that may be impacted by a model trained on this dataset. Are these communities represented in your annotator pool?* The SA-V dataset is a geographically diverse, publicly available, video segmentation dataset, as discussed in 5.2. In addition, we analyze the responsible AI axes of a model trained on the dataset, as discussed in E.1.1.

**Platform and Infrastructure Choices**

1. *What annotation platform did you utilize? At a high level, what considerations informed your decision to choose this platform? Did the chosen platform sufficiently meet the requirements you outlined for annotator pools? Are any aspects not covered?* We used an internal annotation platform.

2. *What, if any, communication channels did your chosen platform offer to facilitate communication with annotators? How did this channel of communication influence the annotation process and/or resulting annotations?* The research team shared feedback daily and met with the annotators weekly to align on the task instructions and expectations and to hold Q&A sessions. Outside of those sessions, annotators had access to a spreadsheet and chat group to facilitate communication with the research team.

3. *How much were annotators compensated? Did you consider any particular pay standards, when determining their compensation? If so, please describe.* (1) The video collecting crowdworkers were compensated with an hourly wage set by the vendor. (2) Annotators were compensated with an hourly wage set by the vendor.

**Dataset Analysis and Evaluation**

1. *How do you define the quality of annotations in your context, and how did you assess the quality in the dataset you constructed?* Annotators were required to follow a training before moving to production queues. Annotators followed a 2-day training session led by the vendor and then were asked to annotate jobs from a training queue. Annotators were able to move from training to production after the vendor Q&A team or the research team reviewed their work and assessed quality. On average, annotators spent 1 - 2 weeks in training before moving to production. Similarly, the vendor and research team Q&A manually reviewed the production queues' annotations daily, sharing feedback daily.

2. *Have you conducted any analysis on disagreement patterns? If so, what analyses did you use and what were the major findings? Did you analyze potential sources of disagreement?* The disagreement patterns were shared daily and weekly during feedback and Q&A sessions.

3. *How do the individual annotator responses relate to the final labels released in the dataset?* The final labels are after data cleaning and post processing from the individual annotator responses.

**Dataset Release and Maintenance**

1. *Do you have reason to believe the annotations in this dataset may change over time? Do you plan to update your dataset?* No.

2. *Are there any conditions or definitions that, if changed, could impact the utility of your dataset?* No.

3. *Will you attempt to track, impose limitations on, or otherwise influence how your dataset is used? If so, how?* The SA-V dataset is released under a permissive CC by 4.0 license.

4. *Were annotators informed about how the data is externalized? If changes to the dataset are made, will they be informed?* No.

5. *Is there a process by which annotators can later choose to withdraw their data from the dataset? If so, please detail.* No.

