# OpenReview forum: "SAM 2: Segment Anything in Images and Videos"
_ICLR.cc/2025/Conference — ICLR 2025 Oral_

### Official Review · Reviewer_yeQ7 · 2024-10-30

**Soundness:** 4
**Presentation:** 4
**Contribution:** 4
**Rating:** 10
**Confidence:** 5

**Summary:**

This paper extends SAM to video, which can segment anything in images and videos. This paper has three significant contributions:
1. The paper expands SAM to video, enabling the segmentation of anything in video.
2. The paper develops a data engine for promptable video segmentation and constructs a large-scale video segmentation dataset, SA-V.
3. The paper designs a more efficient architecture for promptable image and video segmentation, demonstrating significant acceleration.

**Strengths:**

1. The paper is well-written and easy to follow.
2. The paper has significant contributions, including a more efficient model architecture, a large-scale SA-V dataset, and fantastic performance.
3. The paper conducts comprehensive experiments and provides some valuable insights.

**Weaknesses:**

The paper does not have significant weaknesses. My concerns and suggestions are listed in the ``Questions" section.

**Questions:**

1. How would existing SOTA VOS methods perform after finetuning on the SA-V dataset? The paper uses SAM+Cutie as a baseline; however, Cutie has not been trained on large-scale data like SA-V. Therefore, I am interested in Cutie's performance after training with the SA-V dataset. After finetuning with SA-V, would SAM+Cutie surpass SAM2? This would not affect the paper's significant contributions, regardless of the results, as SAM2 is an end-to-end model.

2. I suggest the authors should cite some works from VIS [1, 2, 3, 4], VSS [5], and VPS [6, 7, 8] fields in the related works section, as these areas are also highly relevant to this paper.

[1] Video instance segmentation

[2] End-to-end video instance segmentation with transformers

[3] Tube-Link: A flexible cross tube framework for universal video segmentation

[4] DVIS: Decoupled video instance segmentation framework

[5] Vspw: A large-scale dataset for video scene parsing in the wild

[6] Video-kmax: A simple unified approach for online and near-online video panoptic segmentation

[7] Video k-net: A simple, strong, and unified baseline for video segmentation

[8] OMG-Seg: Is one model good enough for all segmentation?

---

> ### Author Response · Authors · 2024-11-20
> **Response to reviewer yeQ7**
>
> > How would existing SOTA VOS methods perform after finetuning on the SA-V dataset? The paper uses SAM+Cutie as a baseline; however, Cutie has not been trained on large-scale data like SA-V. Therefore, I am interested in Cutie's performance after training with the SA-V dataset. After finetuning with SA-V, would SAM+Cutie surpass SAM2? This would not affect the paper's significant contributions, regardless of the results, as SAM2 is an end-to-end model.
>
> Thank you for your suggestion! We followed your recommendation and fine-tuned the best available Cutie model ("cutie-base-mega") on our SA-V dataset.
>
> After fine-tuning Cutie on the SA-V dataset, the semi-supervised VOS J&F results improved from 62.8 to 69.0 on SA-V test (still largely underperforming the 76.0 J&F obtained by SAM 2 on SA-V test, reported in Table 17a). Also, we found that after fine-tuning, the SAM+Cutie baseline (denoted as "SAM+Cutie (SA-V ft)" in the table below) still largely underperformed SAM 2 on our full benchmark of 17 zero-shot video datasets. Below are the full results:
>
> | model               | 1-click | 3-click | 5-click |  bbox   |  mask   |
> |---------------------|---------|---------|---------|---------|---------|
> | SAM+XMem            |  56.9   |  68.4   |  70.6   |  67.6   |  72.7   |
> | SAM+Cutie           |  56.7   |  70.1   |  72.2   |  69.4   |  74.1   |
> | SAM+Cutie (SA-V ft) |  56.3   |  70.0   |  72.2   |  69.1   |  74.2   |
> | SAM 2               |**64.3** |**73.2** |**75.4** |**72.9** |**77.6** |
>
> *Table: Zero-shot accuracy across 17 video datasets using different prompts (same setting as Table 4 in our submission).*
>
> We are also exploring joint training of Cutie on our SA-V dataset along with previous open-source datasets and will include these results in the final version of the paper.
>
> > I suggest the authors should cite some works from VIS [1, 2, 3, 4], VSS [5], and VPS [6, 7, 8] fields in the related works section, as these areas are also highly relevant to this paper.
>
> We appreciate these suggestions and will ensure that all of these works are cited in the final version of the paper. Thank you for your insightful review. Please let us know if you have any further questions, and we will be happy to address them.

---

> ### Comment · Reviewer_yeQ7 · 2024-11-22
>
> Thanks for the authors thoroughly addressing all of my questions. I decide to raise my score to 10.

---

### Official Review · Reviewer_vraf · 2024-11-01

**Soundness:** 3
**Presentation:** 3
**Contribution:** 3
**Rating:** 8
**Confidence:** 4

**Summary:**

This work addresses promptable visual segmentation in both images and videos. The primary contributions include a new task that generalizes image segmentation to the video domain, a new unified model for video and image segmentation, and a new dataset consisting of 35.5M masks across 50.9K videos. EExtensive evaluations across diverse benchmarks demonstrate that SAM 2 achieves state-of-the-art performance, highlighting its potential to enable "segment anything in videos."

**Strengths:**

* Compared to the original SAM model, SAM 2 improves segmentation accuracy, enabling more precise identification and segmentation of objects in images and videos.
* The processing speed is approximately six times faster than its predecessor. This allows SAM 2 to generate segmentation masks more quickly, making it suitable for real-time applications.
* SAM 2 exhibits strong zero-shot transfer capability.
* The training dataset includes 11 million images and 11 billion masks, providing a robust foundation for new video segmentation tasks for the community.
* The model and dataset are open-sourced.

**Weaknesses:**

From my perspective, there is no obvious weakness in this work. If must to say:
1. The claimed improvement in running speed is mainly due to the usage of the Hiera image encoder, which may not be viewed as a unique contribution of this study.
2. The primary contribution lies in a large-scale dataset and pre-trained models, while the technical contribution is relatively limited.

**Questions:**

N/A

---

> ### Author Response · Authors · 2024-11-20
> **Response to reviewer vraf**
>
> > The claimed improvement in running speed is mainly due to the usage of the Hiera image encoder, which may not be viewed as a unique contribution of this study.
>
> While the use of Hiera is indeed an important factor in improved running speed, we actually use a _simpler_ and _faster_ version of Hiera than prior work (see Appendix D.1 for details on the image encoder). Our version is ~1.5x faster than Bolya et al., ICLR, 2024.
>
> Importantly, our goal is not _just_ improved running speed but improved running speed _while maintaining or improving on accuracy compared to SAM 1 (on images)_. To this end, we found other modeling improvements over SAM 1 to also be important, such as the use of a feature pyramid network in the mask decoder, supervising all IoU predictions, and $l_1$ losses for IoU, etc. (see Appendix D.1 and D.2).
>
> It is the _combination of these modeling changes_ that results in a 6x improvement in running speed (with accuracy >= SAM 1).
>
> > The primary contribution lies in a large-scale dataset and pre-trained models, while the technical contribution is relatively limited.
>
> Our technical contributions are along three axes: **model**, **data engine**, and **task**.
>
> [**Model**] The model is a first-of-its-kind unified architecture for images and videos capable of _real-time inference_ on videos with a step-change in performance. Besides the modeling innovations discussed above (improvements in image encoder and training objectives) which benefit both images and video, our design has several key _video-specific_ innovations: (a) occlusion head + ambiguity handling, (b) memory attention mechanism with spatial and object pointer memories, (c) simple, scalable memory encoder, and (d) support for prompting on any frame. In more detail:
>
>
> - (a) Introduces a way to model ambiguity (and occlusion) in the target object, which is _fundamental_ to the promptable visual segmentation task. Dropping this results in significantly worse quality masks containing visual artifacts. Moreover, the occlusion head allows _principled_ memory selection and can help with longer occlusions (**+2.8 J&F** on SA-V test, see response to reviewer DQTY).
> - (b) We introduce a novel memory attention module with spatial memories and object pointers. Object pointers significantly help performance (**+3.8** on SA-V val), while modeling spatial memories with RoPE allows precise modeling (**+1.2** on SA-V val) of high-resolution spatial information.
> - (c) Contrary to prior work, we do not use a separate image encoder for memories and frame embeddings. This design allows the quality of spatial memories to _scale with the quality of the image encoder_ (unlike prior work). Without this design, we observed worse results from scaling the image encoder in early prototypes.
> - (d) SAM 2 natively supports interactivity on any frame of the video without requiring a separate image segmentation model. Our analysis in Table 1 shows the improvement in annotation speed from a unified SAM 2 compared with a decoupled image segmentation model + tracking model.
>
> [**Data Engine**] The development and training of the SAM 2 model were powered by the availability of a large-scale diverse dataset. However, it is important to note that the developed SAM 2 model also plays a crucial role in the _collection_ of this dataset. SAM 2 was integrated into the annotation tool and could be used _online_ during the annotation task. The model actively contributed to the data collection and refinement, including enabling identification and targeting of annotations on the failure cases of the model. The model advancements and the development of an efficient data engine together enhance the quality and scale of the collected SA-V dataset. This iterative process of model improvement and data collection creates a feedback loop that drives continuous improvement in both the dataset and the model performance. This highlights the technical contributions that underpin the joint development of the large-scale SA-V dataset and SAM 2 model.
>
> [**Task**] We generalize existing tasks by introducing the _promptable visual segmentation task_ for images and videos as well as _relevant online and offline metrics_ to measure performance in this general setting. This task was specifically aligned with our data engine requirements, such that better performance on the task would directly result in a better model for our data collection. Having a clear, general task and relevant metrics is the foundation that made everything else possible.
>
> We appreciate the reviewer's insightful feedback. If there are any other questions or areas where you would like more information, please feel free to reach out, and we will gladly provide further details.

---

### Official Review · Reviewer_DQTY · 2024-11-01

**Soundness:** 4
**Presentation:** 3
**Contribution:** 4
**Rating:** 8
**Confidence:** 4

**Summary:**

This paper proposed a strong foundation model for promptable visual segmentation in images and videos. It proposed a new data engine  that enhances model and data through user interaction, creating the largest video segmentation dataset to date.  A streaming memory augmented transformer is proposed  for real-time video processing.

**Strengths:**

1. This paper proposed a strong foundation model for the video and image segmentation. The data, model, and insights will serve
as a significant milestone for video segmentation.

2. The writing of the paper is good and the paper is easy to understand.

**Weaknesses:**

1. More experiments should be conducted. For example, more interactive VOS methods should be compared.
[1*] Modular interactive video object segmentation: Interaction-to-mask, propagation and difference-aware fusion. CVPR 2021
[2*] Memory aggregation networks for efficient interactive video object segmentation. CVPR 2020

2. More VOS datasets (e.g., VIPOSeg[4*]) should be included in this paper.
[4*] Video Object Segmentation in Panoptic Wild Scenes. IJCAI 2023

**Questions:**

1. How is the annotation quality of the SA-V dataset, and how do you ensure the quality of the annotations? What is the difficulty level of this dataset (such as the movement of objects in the video, occlusions, etc.) compared to previous datasets?

2. Has SAM 2 attempted stability testing for results on ultra-long videos? Is object tracking in long videos more prone to errors?


3. If the memory bank is of fixed size, will it lead to forgetting when dealing with long videos?

---

> ### Author Response · Authors · 2024-11-20
> **Response to reviewer DQTY (1/3)**
>
> > More experiments should be conducted. For example, more interactive VOS methods should be compared. [1*] Modular interactive video object segmentation: Interaction-to-mask, propagation and difference-aware fusion. CVPR 2021 [2*] Memory aggregation networks for efficient interactive video object segmentation. CVPR 2020
>
> Thanks for your feedback! We have also compared with MiVOS [1] in Table 14 of our appendix, under the DAVIS interactive benchmark and following the setting in CiVOS ([A]: [https://arxiv.org/abs/2203.01784](https://arxiv.org/abs/2203.01784)). It was shown that SAM 2 (under click inputs) outperforms MiVOS under the same type of click inputs, and achieves comparable performance to the MiVOS variant under a strong type of scribble inputs. SAM 2 also largely outperforms MANet [2] (which has a lower performance than MiVOS [1]).
>
>
> After our submission, we found that SAM 2 could perform better on this benchmark under slightly different eval parameters. The table below is a summary:
>
>
> | model     | input type | AUC-J&F | J&F@60s |
> |-----------|------------|---------|---------|
> | MANet [2] | scribbles  |  0.79   |  0.79   |
> | MiVOS [1] | scribbles  |  0.87   |  0.88   |
> |-----------|------------|---------|---------|
> | MiVOS [1] | clicks     |  0.75   |  0.75   |
> | CiVOS [A] | clicks     |  0.83   |  0.84   |
> | SAM 2     | clicks     |**0.86** |**0.90** |
>
> *Table: Performance of SAM 2 and other models on the DAVIS interactive benchmark (same setting as Table 14 in our submission).*
>
>
> As noted in appendix F.2 of our submission, we found that SAM 2 often tends to segment object parts (e.g., a person’s arm) on the first click while the DAVIS dataset mainly contains whole objects (e.g., an entire person), which could unfairly penalize SAM 2’s J&F performance on this benchmark for being more flexible.
>
> > More VOS datasets (e.g., VIPOSeg[4*]) should be included in this paper. [4*] Video Object Segmentation in Panoptic Wild Scenes. IJCAI 2023
>
> Thanks, this is a great suggestion! Please note that SAM 2 has been evaluated on 17 zero-shot benchmarks for VOS in Table 4 and Figure 15. Following your suggestion, we also evaluated SAM 2 on the VIPOSeg dataset and found that SAM 2 has a strong performance on this benchmark:
>
> - In a *zero-shot* manner on this dataset, SAM 2 already achieves G=78.4 on VIPOSeg val (where G is the overall metric defined in [4], higher is better), on par with the G=78.2 performance of PAOT (the best model in Table 2 of [4] *trained* on this dataset).
>
> - When fine-tuned on the VIPOSeg training set, SAM 2's performance is further improved to G=79.7 on VIPOSeg val.
>
> - In addition, SAM 2 is more robust in crowded scenes of VIPOSeg as measured by the decay metric defined in [4] (lower is better). On VIPOSeg val set, SAM 2 achieves λ=0.68 decay (without fine-tuning on VIPOSeg) and λ=0.67 decay (when fine-tuned on VIPOSeg train set), both outperforming the λ=0.70 decay of PAOT (the best model in Table 2 of [4] trained on this dataset).
>
> We will add these results on VIPOSeg to the final version of the paper.

---

> ### Author Response · Authors · 2024-11-20
> **Response to reviewer DQTY (2/3)**
>
> > How is the annotation quality of the SA-V dataset, and how do you ensure the quality of the annotations? What is the difficulty level of this dataset (such as the movement of objects in the video, occlusions, etc.) compared to previous datasets?
>
> To ensure annotation quality we include a rigorous quality verification step in the data engine (detailed in Section 5.1). A separate set of annotators are tasked with verifying the quality of each annotated masklet, ensuring _every_ masklet in the SA-V dataset is "satisfactory", i.e., it correctly and consistently tracks the target object across all frames. Unsatisfactory masklets were sent back to the annotation pipeline for refinement.
>
> We use the disappearance rate (the percentage of annotated masklets that disappear in at least one frame and re-appear) as the proxy for occlusion and the difficulty level of datasets. The SA-V manual subset has the highest disappearance rate (42.5%) compared with all previous datasets. The comparison of SA-V with respect to previous datasets on dataset size, video duration, and disappearance rate is provided in Table 3.
>
> We compare the mask size distribution, normalized by video resolution, with previous datasets including DAVIS, MOSE, and YouTubeVOS, in Figure 10 (a). More than 88% of SA-V masks have a normalized mask area less than 0.1, and the SA-V dataset contains more small objects compared to MOSE and YouTubeVOS. In addition, the SA-V dataset also contains annotations for object parts as well as whole objects, compared to previous datasets which focus on whole objects across common categories.
>
> We also find the SA-V benchmark to be challenging in terms of absolute J&F accuracy. As summarized in the table below (from Table 6 in the paper), the J&F accuracy of SAM 2 models is lower on SA-V val compared to DAVIS 2017 val and YouTubeVOS 2019 val. This suggests that the SA-V benchmark is a challenging testbed.
>
> | J&F              | SA-V val | MOSE val | DAVIS 2017 val | LVOS val | YouTubeVOS 2019 val |
> |------------------|:--------:|:--------:|:--------------:|:--------:|:-------------------:|
> | SAM 2 (Hiera B+) |   73.6%  |   75.8%  |      90.9%     |   74.9%  |        88.4%        |
> | SAM 2 (Hiera L)  |   75.6%  |   77.2%  |      91.6%     |   76.1%  |        89.1%        |

---

> ### Author Response · Authors · 2024-11-20
> **Response to reviewer DQTY (3/3)**
>
> > Has SAM 2 attempted stability testing for results on ultra-long videos? Is object tracking in long videos more prone to errors?
>
> Thanks for raising this question. Longer videos have a larger discrepancy from the training setting, and it is expected there will be some performance degradation (_vs_ shorter videos). Indeed, we can see such a degradation below where we bin videos by duration and report performance on these bins.
>
> While SAM 2 was not specifically designed for long videos (an important and challenging research direction on its own), it nevertheless shows a _step-change_ in performance on long VOS benchmarks such as LVOS v1 and v2 (e.g. > **+16 J&F** on LVOS v2 for SAM 2.1 (L) vs previous best reported performance).
>
> Moreover, with some further modeling improvements, we are able to mitigate degradation on long videos, see table below where we measure the mean (SEM) performance across varying video durations of the LVOSv2 dataset. Notice that the _gains are larger for longer videos_.
>
> | **Model** 	| **0-50 sec** 	| **50-100 sec** 	| **> 100 sec** 	|
> |---|---|---|---|
> | SAM 2 	| 84.4 (0.02) 	| 79.9 (0.03) 	| 75.9 (0.04) 	|
> | SAM 2.1 	| 85.5 (0.02) 	| 82.9 (0.02) 	| 79.0 (0.03) 	|
>
> *Table reports performance on the semi-supervised VOS task on LVOS v2, binned by video duration.*
>
> Finally, note that since SAM 2 is an _interactive_ model, it also allows corrections in a later frame to recover performance.
>
> > If the memory bank is of fixed size, will it lead to forgetting when dealing with long videos?
>
> This is a good point that warrants further discussion. The memory bank for SAM 2 indeed has a fixed maximum capacity. However, this is also the case with nearly all practical models, from prior work in video object segmentation to LLMs (due to their finite budget of e.g., GPU memory). Prior works often use various ad-hoc heuristics (e.g., [1]) to decide how to use the limited model capacity effectively.
>
> In contrast with ad-hoc approaches, SAM 2 is *carefully designed to mitigate such issues*:
>
> - By default SAM 2 _always_ retains *all prompted frames* in the memory bank, which allows it to remember the prompted information regardless of the video length and alleviates forgetting.
>
> - SAM 2 is an _interactive model_ and can therefore be _prompted_ on any frames in the video. So even when SAM 2 makes a mistake in long videos, it can be easily fixed with additional prompting.
>
> - SAM 2’s _occlusion head_ (which predicts whether the object being tracked is present on a frame) allows a _principled_ way to select memories to be placed/retained in the memory bank. Adopting such a strategy further improves the performance, e.g., we find SAM 2.1’s performance improves by **+2.8 J&F** from 78.2 to 81.0 on SA-V test in the semi-supervised VOS setting.
>
> - Finally, SAM 2 has two kinds of memories, spatial memories and _object pointer_ memories - the latter are $\approx10^3$ times _smaller_ than spatial memories, which allows us to retain them over longer temporal context windows and further mitigates forgetting in long videos.
>
> [1] A Simple and Effective L2 Norm-Based Strategy for KV Cache Compression, Devoto et al, EMNLP 2024.
>
> We thank the reviewer for their insightful review and suggestions for improvement. Please let us know if anything else is unclear, and we are happy to answer.

---

> > ### Comment · Reviewer_DQTY · 2024-11-27
> >
> > The authors addressed all my concerns and I will keep my score.

---

### Official Review · Reviewer_DAoa · 2024-11-02

**Soundness:** 4
**Presentation:** 4
**Contribution:** 3
**Rating:** 10
**Confidence:** 4

**Summary:**

In this paper, the authors build a data engine to generate a large-scale video segmentation dataset. Using the datasets, they train a strong yet efficient model.

**Strengths:**

1. With the data engine pipeline, the paper provides an extremely large-scale video segmentation dataset compared to previous datasets. This will allow the researchers to tackle much more challenging tasks in video segmentation.

2. Based on the experimental results, the trained SAM2 model outperforms the combination of SAM and existing state-of-the-art trackers by a large margin. Therefore, the data scaling-up with the data engine is effective, as described by the authors. The results also imply the potential of further data scaling up with the data engine.

3. For image segmentation, the SAM2 model can also perform better, even with a much smaller computational cost. This will facilitate the applications of the SAM2 model. In a constrained platform, it is always better to have a good and efficient model.

4. The paper provides detailed information about the implementations of the data engine and data distribution. It is also good that the authors release their data and models. It would be even better if the training code and data engine could be publicly available.

**Weaknesses:**

1. Although this paper uses a simpler structure and performs well, it is still possible to use previous structures, such as Cutie [R1], to achieve even better performance with SAM2 data. It would be better if the structure could be explored.

2. SAM2 cannot recognize segmented objects like previous models [R2, R3]. It would be better to discuss this since it may limit the application of this paper. It would also be better to discuss the difference with [R4], which supports image and video segmentation and can recognize the objects.


[R1] Putting the Object Back into Video Object Segmentation.

[R2] Open-Vocabulary SAM: Segment and Recognize Twenty-thousand Classes Interactively.

[R3] Semantic-SAM: Segment and Recognize Anything at Any Granularity.

[R4] OMG-Seg: Is One Model Good Enough For All Segmentation?

**Questions:**

NA

---

> ### Author Response · Authors · 2024-11-20
> **Response to reviewer DAoa**
>
> > It would be even better if the training code and data engine could be publicly available.
>
> We agree. We are releasing training code and the code for a web demo which resembles our annotation tool for the SAM 2 data engine. In Appendix E2 we provide further details on the annotation protocol and tasks (Figure 11). We are also releasing inference and fine-tuning code, together with the models under a permissive license.
>
> > Although this paper uses a simpler structure and performs well, it is still possible to use previous structures, such as Cutie [R1], to achieve even better performance with SAM2 data. It would be better if the structure could be explored.
>
> This is a valuable discussion point. The Cutie model is specifically designed for the _much narrower_ semi-supervised VOS task for _videos_, whereas the SAM 2 model is intended for *broader capabilities* and is *capable of interactive promptable segmentation on images and videos*.
>
> That said, we anticipate that using our SA-V dataset to train VOS models (including Cutie) should yield significant improvements.
>
> Following your suggestion, we conducted an experiment to evaluate this.
>
> We implemented a custom SA-V dataloader in the Cutie codebase to train it on our SA-V dataset. After training on SA-V, Cutie achieves 69.0 J&F on SA-V-test compared to 62.8 J&F for the official Cutie checkpoint .
>
> This 6.2-point boost shows the impact of our large-scale SA-V dataset in enabling the "track anything" capability, as evaluated by the SA-V benchmark. Although Cutie's performance on SA-V-test greatly benefits from this SA-V training data, it still falls short of the 76.0 J&F achieved by SAM 2 (Table 17a).
>
> We observed that purely training on SA-V degrades the performance of Cutie on benchmarks that have their own in-domain training sets, MOSE and DAVIS, which become zero-shot if not training on these. We are currently experimenting with jointly training Cutie on our SA-V dataset as well as previous open-source datasets and will report the results as soon as they come available and will report them in the final version of the paper.
>
> > SAM2 cannot recognize segmented objects like previous models [R2, R3]. It would be better to discuss this since it may limit the application of this paper. It would also be better to discuss the difference with [R4], which supports image and video segmentation and can recognize the objects.
>
> Thank you for the pointers! While these are indeed interesting works, our focus is on the *promptable visual segmentation task* (PVS), and therefore, recognition is beyond the scope of this current work.
>
> However, the progress we have made on the PVS task can certainly facilitate significant advances in recognition! Indeed, R2, R3, and R4 were built upon SAM 1 (or the SA-1B dataset) and delivered advances in recognition. We believe that SAM 2 can enable similar advances for (images and) _video_.
>
> Please note that among these works (R2, R3, and R4), only R4 has video capability, and SAM 2 significantly outperforms it on the VOS task (e.g., [R4] has only 76.9 J&F on DAVIS 2017 _vs_ 90.9 J&F of SAM 2). As such, these works are complementary to SAM 2, and we will discuss them in the paper.
>
> We would like to thank the reviewer for their insightful review. Please let us know if there are any other questions or points you would like us to address, and we will be happy to do so.

---

> ### Comment · Reviewer_DAoa · 2024-11-23
>
> Thanks for the authors' comments on the weaknesses I previously mentioned. I would like to keep the score at 8, and I also look forward to the experiments mentioned by the authors in their response. It is a good paper.

---

> ### Author Response · Authors · 2024-11-27
> **Update on the completed experiments for reviewer DAoa**
>
> All the experiments mentioned in our first response have finished now.
> Following your suggestion, we aim to evaluate if the SA-V data can also benefit existing models, beyond SAM 2. To evaluate this, we use SA-V to train a recent semi-supervised VOS model, Cutie [1], which SA-V was *not* collected for, to assess SA-V's generalization capabilities. For this, we implemented a custom dataloader to jointly train Cutie on DAVIS, YouTubeVOS, MOSE and our SA-V dataset.
>
> To assess the generality of SA-V, we compare the performance of Cutie models trained with and without it, both for the existing VOS task with mask or box prompt on the first frame, and interactive video segmentation, identical to the setup in §6.2. We report results in the following table.
>
> | Method        | SA-V test | MOSE val | DAVIS 2017 test | LVOS val | 5-click | box  | mask |
> |---------------|-----------|----------|-----------------|----------|---------|------|------|
> | Cutie         | 62.2      | 68.3     | 85.3            | 63.5     | 71.7    | 68.6 | 73.5 |
> | Cutie + SA-V  | 68.7      | 67.6     | 84.5            | 65.0     | 73.4    | 70.4 | 75.5 |
> | SAM 2.1         | **78.4**  | **77.9** | **87.7**        | **78.0** | **77.6**| **74.4** | **79.3** |
>
> **Table:** Our SA-V data has a broader utility beyond SAM 2 and helps zero-shot generalization in other models. Including SA-V in the training mix improves the performance of Cutie on SA-V test and provides zero-shot gains on LVOS and 17 zero-shot datasets. The final performance of Cutie + SA-V still falls short of the full SAM 2 shown in the last row.
>
> As a baseline, the first row of the table shows the J&amp;F accuracy of the official Cutie model trained on DAVIS, YouTubeVOS, and MOSE (referred to as "w/ MOSE" in [1]) as evaluated by us on SA-V test and our zero-shot benchmark of 17 datasets (using 5 clicks or bounding box as input to SAM+Cutie or ground-truth mask input to Cutie; see §F.1.3  for details). Performance on MOSE, DAVIS, and LVOS benchmarks is from officially reported numbers.
>
> In the second row of the table, we train another Cutie model on our data mixture consisting of SA-V, DAVIS, YouTube-VOS, and MOSE. We use the same data mixture ratio as for SAM 2 and follow the official Cutie training recipe.
>
> Including SA-V in the training mix brings a large performance improvement on the SA-V benchmark. Although integrating SA-V into the training mix does not improve the performance on MOSE and DAVIS *in-domain* benchmarks, it boosts the *zero-shot* performance as shown by gains of +1.5 on LVOS and +2.0 on our 17 zero-shot benchmark suite under mask input, and similar boost for the out-of-domain click and box prompts.
>
> Finally, the last row shows the accuracy of SAM 2. Note that SAM 2 significantly outperforms the Cutie models under both training settings, suggesting that the SAM 2 architecture design is another crucial aspect of the final performance.
>
> Thank you for your great suggestion, this really makes the paper stronger and shows the generalization of the data beyond the SAM 2 model architecture.
>
> [1] Putting the Object Back into Video Object Segmentation, Cheng et al, CVPR 2024

---

### Author Response · Authors · 2024-11-20
**Author response summary**

We thank all reviewers for their comprehensive reviews and insightful suggestions. We are pleased to receive numerous positive remarks, such as:

- [DAoa]: "Provides an extremely large-scale video segmentation dataset...will enable researchers to tackle more challenging tasks," "outperforms existing state-of-the-art trackers by a large margin," "good and efficient model."

- [DQTY]: "A strong foundational model for video and image segmentation," "_data, model, and insights will serve as a significant milestone_"

- [vraf]: "Extensive evaluations demonstrate that SAM 2 achieves state-of-the-art performance, highlighting its potential to enable 'segment anything in videos,'" "exhibits strong zero-shot transfer capability," "six times faster than its predecessor," "_no obvious weakness_,"

- [yeQ7]: "Significant contributions, including a more efficient model architecture, a large-scale SA-V dataset, and fantastic performance," "comprehensive experiments provide valuable insights," "well-written and easy to follow," "_does not have significant weaknesses_."

Following our initial submission, we have made further enhancements to SAM 2 through model advancements, below referred to as "SAM 2.1." These improvements not only enhance overall performance, but also address specific reviewer inquiries, such as reviewer DQTY's question regarding performance on longer videos, where SAM 2.1 demonstrates notable gains in LVOS v2.

| Model                        | SA-V Test | MOSE | LVOS v2 |
|------------------------------|-----------|------|---------|
| SAM 2 (Hiera-B+)             | 74.7      | 72.8 | 75.8    |
| SAM 2.1 (Hiera-B+)           | 78.2      | 73.7 | 78.2    |
| SAM 2 (Hiera-L)              | 76.0      | 74.6 | 79.8    |
| SAM 2.1 (Hiera-L)            | 79.5      | 74.6 | 80.6    |

*Table shows J&F metric on the semi-supervised VOS task. Training data included SA-1B, SA-V, and Internal-train. We will release all these models and incorporate these results into the paper.*

**Technical Details for These Improvements:**

(a) We enhanced SAM 2's occlusion handling capability and stability over longer durations adding a no-object embedding in spatial memories and temporal position encoding in object pointers, as well as by training the model on extended sequences of frames (16 frames) in an additional training stage.

(b) We introduced mosaic data augmentation to simulate the presence of visually similar and small objects.

We are revising the draft to include information on these improvements.

Please see below for our responses to the individual reviewer questions.

---

> ### Author Response · Authors · 2024-11-23
> **Author response summary**
>
> Dear reviewers, we have updated the paper draft to incorporate these SAM 2.1 enhancements, along with the updates provided in the rebuttal.

---

> > ### Author Response · Authors · 2024-11-27
> > **Author response summary**
> >
> > Dear reviewers, we have completed a final revision of the paper draft. In response to reviewer DAoa's suggestion, we have added a section discussing the benefits of training the Cutie model on SA-V for zero-shot performance. You can find this information in Section G of the appendix. Thank you once again for your time and effort in providing insightful reviews for our paper.

---

### Meta-Review · Area_Chair_jgxA · 2024-12-15

**Metareview:**

This work presents a visual foundation model, SAM 2, achieving new state-of-the-art results on video/image interactive segmentation and video object segmentation.

Several issues in the earlier version are about missing compared methods, close related works on video semantic-level segmentation (video instance segmentation, video panoptic segmentation), or other vision foundation models.

All these issues are solved after rebuttal and discussions between reviewers and authors. Thus, all reviewers accept this work, including two strong accepts.

SAM 2 will be a strong baseline and a product-level model for the AI community. Thus, the Area Chair suggests accepting this work as an oral presentation.

**Additional Comments On Reviewer Discussion:**

All the issues raised by reviewers are well solved.

---

### Decision · Program_Chairs · 2025-01-22

Accept (Oral)